# LAYER QUERY NETWORK FOR TEST-TIME-TRAINING IN VISION-LANGUAGE-MODELS

## ABSTRACT

Vision–Language Models (VLMs) struggle to generalize against out-of-distribution (OOD) samples, where conventional fine-tuning is infeasible. *Test-Time Training (TTT)* adapts models to each incoming test sample, yet current methods rely on heavy data augmentation and repeated forward/backward passes through the full VLM, incurring high computational cost. We introduce **Layer Query Network (LQN)**, a lightweight five-layer MLP that adapts a frozen VLM in *one forward pass*. LQN employs *Binding* to distill randomly sampled intermediate-layer tokens from VLM via 3D positional embeddings, and *Recirculation* to self-supervise spatial invariance for predicting robust spatially consistent features. This design removes the need to fine-tune the entire VLM, achieving faster convergence and strong dense-prediction performance, outperforming the teacher VLM. Evaluated across 16 benchmarks spanning natural distribution shifts and cross-dataset generalization, LQN achieves $15\%$ faster test-time training on ImageNet-Val compared to the state-of-the-art TPS. In segmentation tasks, LQN surpasses Mask2Former on COCO, Cityscapes, and ADE20K while reducing GFLOPs by up to $11\%$. Our code will be released upon acceptance.

## 1 INTRODUCTION

CLIP (Radford et al., 2021) is widely regarded as one of the first Vision–Language Model (VLM) to demonstrate strong zero-shot generalization in tasks like image classification and image–text retrieval. Its success has spawned a variety of downstream applications like image segmentation (Wang et al., 2025), video text retrieval (Hur et al., 2025), audio classification (Dixit et al., 2024), *etc*. As real-world deployment of VLMs grows, the key question emerges: *"Is there an efficient way to boost the out-of-distribution generalization of VLM-based systems for real-world?"*

VLMs systems are known to show sub-optimal performance under distribution shift, like unseen test domain / **O**ut-**o**f-**d**istribution (**OOD**) (Shu et al., 2023; Mayilvahanan et al., 2023). A significant effort in Computer Vision explores fine-tuning methods, such as Adapters (Yin et al., 2023), LoRA (Hu et al., 2022), and VPT (Jia et al., 2022) *etc*. that can adapt models to new datasets while retaining generalization. However, these approaches assume access to the labeled target dataset, an assumption that rarely holds in real-world deployments, limiting their practicality.

To improve generalization, *Test-Time Adaptation (TTA)* methods adapt models by "peeking" at target data for on-the-fly domain adjustment (Zhang et al., 2024; Osowiechi et al., 2024). Most TTA approaches, however, rely on multiple test samples (or a cache of past context) to progressively refine predictions (Nguyen et al., 2025; Karmanov et al., 2024), an assumption that breaks down in data-constrained scenarios such as medical diagnostics, where only a single test instance may be available. To overcome this limitation, a more constrained **T**est-**T**ime **T**raining (**TTT**) has emerged, adapting a model using **just one test sample** (Kojima et al., 2025).

Test-Time Training (TTT) methods such as TPT (Shu et al., 2022) and TPS (Sui et al., 2025) augment each test sample and enforce prediction consistency across those augmentations for unsupervised prompt fine-tuning (fig. 1 (a)). Although effective, they demand: i) *Multiple augmentations*, optimal augmentation pipeline needs to be known beforehand ii) *High computational cost* : every augmentation requires image-encoder forward passes. To reduce these overhead, APM (Modi & Rawat, 2024a) distills a lightweight student from a frozen VLM teacher with just one forward pass (fig. 1 (b)), avoiding repeated VLM forward / backward passes. However, it faces two key issues:

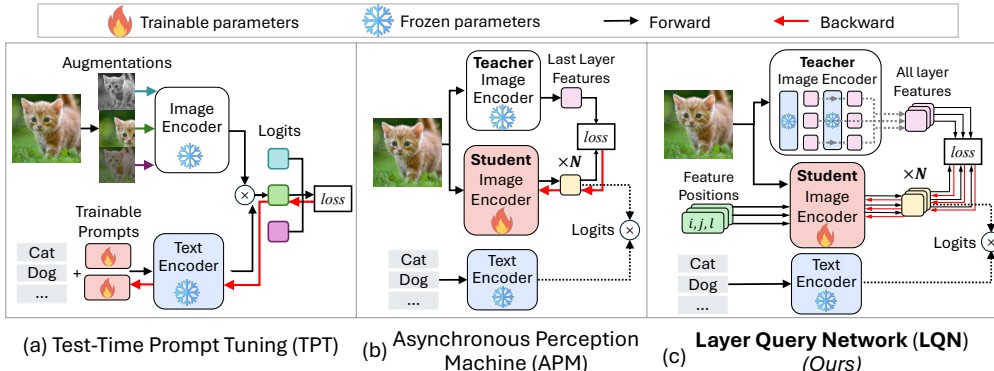

Figure 1: **Comparison with existing work:** *(a) Left:* TPT applies consistency across multiple augmentations to train trainable prompts via text encoder, requiring backprop through the text backbone. *(b) middle:* APM removes the constraint of multiple augmentations and text backbone, by iteratively distilling over the image encoder. *(c) right:* Our method introduces positional encoding (spatial and layer awareness) to APM, and distills intermediate features enabling dense task predictions.

**i) Shallow distillation:** Learning only from final-layer features misses fine-grained nuances, restricting use to sparse tasks such as image classification. **ii) Slow convergence:** It still needs many iterations to match or surpass the teacher's performance, adding to the computation cost of test time.

Our design aims to answer *"Does the entire VLM need to be fine-tuned to handle a single OOD test sample?"* To explore this, we introduce **L**ayer **Q**uery **N**etwork (**LQN**), a lightweight five-layer MLP that efficiently adapts a VLM system for OOD generalization (fig. 1 (c)). Building on APM's advantages, LQN avoids both multiple data augmentations (and their repeated forward passes) and backpropagation through the text encoder. The LQN framework employs two core strategies via the proposed *3D-binded algorithm*: **1) Binding** extends shallow distillation beyond the teacher's final layer output by randomly sampling intermediate teacher-layer tokens, queried with 3D positional embeddings. **2) Recirculation** is a self-supervised step that enforces spatial invariance, enabling LQN to produce more robust, spatially consistent features. This yields two main benefits: **i) Faster convergence**: fewer training iterations and reduced GFLOPs, whilst surpassing teacher VLMs. **ii) Better dense predictions**: Intermediate features improve spatial understanding, boosting dense prediction tasks like image segmentation.

In summary, we present **L**ayer **Q**uery **N**etwork (**LQN**), a lightweight MLP for efficient test-time training and on-the-fly adaptation. LQN converges quickly, reducing GFLOPs while outperforming its teacher VLM in zero-shot and out-of-distribution dense tasks such as image segmentation. We evaluate LQN on 16 benchmarks covering natural distribution shifts and cross-dataset generalization. Key results include: 15% faster test-time training on ImageNet-val compared to the state-of-the-art TPS. Strong segmentation performance on COCO, Cityscapes, and ADE20K, surpassing Mask2Former while cutting GFLOPs by up to 11%. These results demonstrate LQN's superior efficiency and robust test-time generalization.

## 2 RELATED WORK

**Adapting VLMs via Fine-tuning:** Text prompt augmentations were deployed by CLIP to achieve strong zero-shot image classification. Descriptive prompts crafted via LLMs (Pratt et al., 2023; Ren et al., 2023) have been shown to improve adaptation. Inspired by parameter-efficient transfer learning (Lester et al., 2021; Houlsby et al., 2019), follow-up methods improve CLIP adaptation using adapters (Gao et al., 2023) and cross-modal adaptation (Lin et al., 2023).

**Test-Time-Optimization**: consists of both Test-Time Adaptation (TTA)/ Test-Time Training (TTT) approaches. TTA often requires access to *multiple* test samples *simultaneously* to progressively achieve stable adaptation and refined prediction (Wang et al., 2020; Liu et al., 2021; Prabhudesai et al., 2023; Wang et al., 2022; Yuan et al., 2023a; Gong et al., 2022; Yuan et al., 2023b; Gong et al., 2024). Test-Time Training (TTT) approaches like TPT (Shu et al., 2022) enhances test-time robustness via enforcing consistency across augmented views. Prompt-tuning learns trainable textual-prompts by conditioning on input features (Zhou et al., 2022b;a). While effective, forward/backward passes through the text encoder makes it costly. TPS (Sui et al., 2025) speeds this up by adjusting

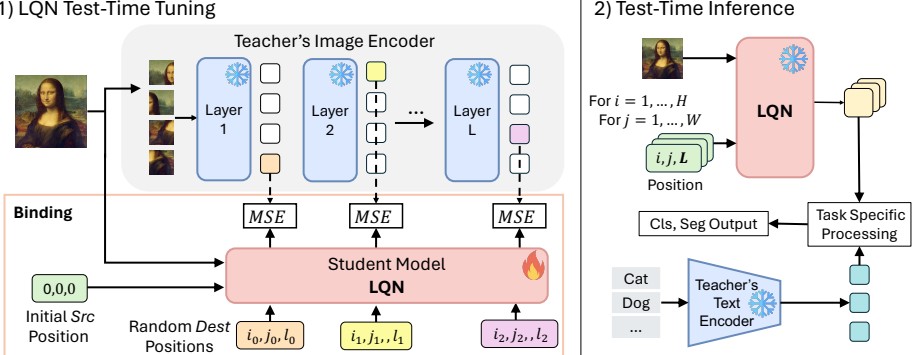

Figure 2: **Layer Query Networks (LQN).** For a single test image, a *frozen* teacher extracts intermediate layers' feature across all layers for supervision. During *test-time tuning*, LQN runs for $N$ iterations by randomly sampling different destination positions $(i, j, l)$, where $(i, j)$ is the spatial location and $l$ the layer depth. The binding procedure takes the image, $(0, 0, 0)$ as the *Src* position, and the sampled *Dest* as input, and the LQN is trained to predict the teacher's corresponding feature via MSE loss. During *test-time inference*, after the final iteration, LQN is queried at all spatial locations of the last layer $L$, and the resulting features are processed by task-specific modules, such as CLIP's textual encoder for zero-shot classification or the teacher's segmentation head for dense prediction.

pre-computed vectors in the feature-space instead of back-propogating through text encoder. Recently, MTA Zanella & Ben Ayed (2024) leverages a mean-shift test-time augmentation approach, which performs unsupervised inlier-score modulation across augmented views. Similarly, GS-Bias Huang et al. (2025) adds global and spatial biases to the logits of a base-model.

Other approaches synthetically *generate* out-of-distribution test data using models like Stable Diffusion (Rombach et al., 2022). Yet almost all TTT methods still incur heavy computation cost from extensive data augmentation, where each augmentation requires forward pass through image encoder and increasing compute overhead. Methods like TTT-MAE enhance model adaptation by introducing self-supervised task, such as image rotation prediction or masked reconstruction. Our LQN follows the experimental-setup in Modi & Rawat (2024a;b), i.e. processing 1 sample at a time, without requiring dataset-specific pre-training or more than one instance. Inspired by APM (Modi & Rawat, 2024a), our LQN *does not* tune the text-encoder at all, avoiding high computation cost.

## 3 METHOD

We introduce the *Layer Query Network (LQN)*, a test-time training framework designed to enhance the out-of-distribution generalization of vision-language models (VLMs). LQN relies on a distillation-based approach and queries and distills spatial tokens across *all* layers of the teacher VLM. By modeling directional relations between pairs of spatial-depth locations, LQN learns to bind target representations and enforce spatial invariance through a novel binding–recirculation procedure. This enables a lightweight student network to recover rich hierarchical representations from the teacher, ultimately improving zero-shot classification and segmentation without requiring additional supervision or modifications to the frozen teacher model. First we describe the *Preliminaries* (section 3.1) to give background on CLIP and the APM algorithm, and then dive deeper into Layer Query Network & 3D-Binded Algorithm (section 3.2).

### 3.1 PRELIMINARIES

**Zero-Shot CLIP:** A VLM like CLIP (Radford et al., 2021) is typically trained on millions of (image,text) pairs. It consists of two parallel encoders, an image encoder $\mathcal{T}_{img}$ and a text-encoder $\mathcal{T}_{text}$. Given a test image $x_{ood}$, and a class label description $cls$, corresponding features are produced as:

$$Y_{img} = \mathcal{T}_{img}(x_{ood}), \quad Y_{text} = \mathcal{T}_{text}(cls), \quad \mathbb{P}_{\text{CLIP}}(y = y_c \mid x) = \frac{\exp(Y_{text_c} \cdot Y_{img})/t}{\sum_{c'} \exp(Y_c \cdot Y_{img})/t}, \quad (1)$$

where, $Y_{img}$ and $Y_{text}$ are vision and text embeddings, respectively, $t$ is the temperature parameter in softmax and $\mathbb{P}_{\text{CLIP}}(y = y_c \mid x_{ood})$ is the probability logit corresponding to this class label $T_c$. The max probability logit over $C$ class is chosen as class prediction.

**Test-Time Training (TTT) setup:** Following recent work, APM (Modi & Rawat, 2024a) (shown in algorithm 2), we adopt a teacher student distillation framework (fig. 1(b)). The teacher is a frozen VLM (*e.g.* CLIP, ViT-L), while the student $S$ is *optimized* at test time for $N$ iterations. First the teacher produces text and images embedding corresponding to $x_{ood}$ and class label $cls$, *i.e.* $Y_{img}$, and $Y_{text}$ (eq. (1)). For $N$ iterations, the students randomly select a batch of spatial indices (i,j) and tries to *mimic* teacher embeddings on those spatial indices. The student $S$ uses positional embedding (Vaswani et al., 2017) and the input image $x_{ood}$ to generate image features. MSE loss is used to train the student. After $N$ iterations, the student uses the teacher text encoder to generate the class logit. It has been observed that such light-weight students can inherit (& surpass) teacher's zero-shot generalization on OOD samples. We reset the students after $N$ iterations, to prevent information-leakage across test-samples.

## 3.2 LAYER QUERY NETWORK (LQN)

LQN student $S_{img}$ adapts a similar setup as APM. Instead of just distilling on the final layer features $Y_{img}$, here we collect spatial tokens across *all the layers*, $Y_{img}^{1,2,...L}$ where $L$ denotes all the layers of teacher VLM. It can be interpreted as a collection of $d$-dimensional vectors defined over a 3D grid of spatial and depth locations, totaling $H \times W \times L$ positions. For the input test sample $x_{ood}$, the student $S_{img}$ mimics the teacher's (i,j) spatial token on depth $l$ *i.e.* $S_{img}(x_{ood}, i, j, l) \rightarrow f$, where $f \in \mathbb{R}^d$ is the predicted vector at position $(i, j, l)$. We can query $S_{img}$ with different positions $(i, j, l)$ in parallel. Previously, it has been observed that directly feeding a 3D position $(i, j, l)$ as an integer to a neural network leads to poor convergence (Mildenhall et al., 2021). Therefore, we encode location $(i, j, l)$ as a 3D positional-encoding $P(i_d, j_d, l_d)$, similar to transformers (Vaswani et al., 2017).

---

**Algorithm 1 The 3D-Binded Algorithm**: Layer Query Network(inspired by algorithm 2).

---

**Input**: Teacher Image/Text Encoder $\mathcal{T}_{img}$ / $\mathcal{T}_{text}$, Student $\mathcal{S}_{img}$, $N$ iterations

**Require:** OOD image $x_{ood}$, Class Label $cls$ & Predict class logit

1: // Teacher **all layers** image features and text feats
2: $Y_{img}^{1,2...L} \leftarrow \mathcal{T}_{img}(x_{ood}) \in \mathbb{R}^{H \times W \times L \times D}$
3: $Y_{text} \leftarrow \mathcal{T}_{text}(cls)$
4: $\mathcal{S}_{img} \leftarrow 0$             // Initialize the student weight
5: **for** iteration $k \in N$ **do**

6:     **# Binding**
7:     $i_d, j_d, l_d \leftarrow \text{Sample(H,W,L)}$          // 'Dest' random spatial + layer index
8:     $P_{i_d,j_d,l_d} \leftarrow P(i_d, j_d, l_d)$          // Dest Position embedding
9:     $\text{Dest} \leftarrow [P_{i_d,j_d,l_d}|P^T[1]]$       //Indicate $(i_d, j_d, l_d)$ is 'dest' by concatenating with flag $P^T[1]$
10:    $L_B \leftarrow \|\mathcal{S}_{img}(\text{Dest}, x_{ood}) - Y_{img}[i_d, j_d]\|_2^2$      // $S_{img}$ predicts 'Dest' features

11:    **# Recirculation**
12:    $i_1, j_1, l_1 \leftarrow \text{Sample(H,W,L)}$          // src 1 random spatial index
13:    $P_{i1,j1,l1} \leftarrow P(i_1, j_1, l_1)$          // Sample Position embedding for src 1
14:    $\text{Src1} \leftarrow [P_{i_1,j_1,k_1}|P^T[0]]$       //Indicate $(i_1, j_1, l_1)$ is 'src1' by concatenating with flag $P^T[0]$
15:    $i_2, j_2, l_2 \leftarrow \text{Sample(H,W,L)}$          // src 2 random spatial index
16:    $P_{i2,j2,l2} \leftarrow P(i_2, j_2, l_2)$          // Sample Position embedding for src2
17:    $\text{Src2} \leftarrow [P_{i_2,j_2,k_2}|P^T[0]]$       //Indicate $(i_2, j_2, l_2)$ is 'src2' by concatenating with flag $P^T[0]$
18:    // $S_{img}$ self-supervises to predict 'Dest' features
19:    $L_R \leftarrow \|\mathcal{S}_{img}(Src1, Dest, x_{ood}) - \mathcal{S}_{img}(Src2, Dest, x_{ood})\|_2^2$

20:    **# Update using both losses**
21:    $\text{Loss} \leftarrow L_B + \alpha L_R$
22:    Update $\mathcal{S}_{img}$
23: **end for**
24: **for** $\forall(i, j) \in (H, W)$, last layer $L$ **do**
25:    $P_{i,j,l} \leftarrow P(i, j, L)$
26:    $\text{Src} \leftarrow [P_{0,0,0}|P^T[0]]$          // Constant source $(0, 0, 0)$
27:    $\text{Dest} \leftarrow [P_{i,j,l}|P^T[1]]$       //Indicate $(i, j, l)$ is 'dest' by concatenating with flag $P^T[1]$
28:    $Y_{img}^{Student} \mathrel{+}= \mathcal{S}_{img}(Src, Dest, x_{ood}) / (H \cdot W)$      //Spatial average of $x_{ood}$ via $S_{img}$
29: **end for**
30: $P_{cls} \leftarrow Y_{img}^{Student} \cdot Y_{text}$          Teacher Text + Student image produces logit
31: **Output:** $P_{cls}$

---

### 3.2.1 LQN MODEL ARCHITECTURE

As opposed to a single spatial relation, LQN student $S_{img}$ operates on a *pair* of locations $(src, dest)$, and encodes a *directional relation* between them. This can be formulated as $S(src, dest, x_{ood})$. For example, if the student $S_{img}$ operates on $(src, dest)$, it should mimic the teacher's representation at *dest*. Consider two *randomly sampled* locations $src = (i_s, j_s, l_s)$, and $dest = (i_d, j_d, l_d)$. The student should be able to *distinguish* between which location is the $src$ and which location is the $dest$. As shown in algorithm 1, we distinguish between a pair of locations $(src, dest)$ by an additional 'flag positional-encoding' $P^T \in \mathbb{R}^{2 \times D}$. $P^T[0]$ indicates which location is the 'source', whereas $P^T[1]$ indicates which is the 'destination'. We generate $(src, dest)$ as:

$$Src = [\, P_{i_s, j_s, l_s} \mid P^T[0]\,], \quad Dest = [\, P_{i_d, j_d, l_d} \mid P^T[1]\,] \qquad \text{(lines 8, 10 in algorithm 1)} \tag{2}$$

where $\mid$ denotes the concatenation operator, $P_{i_s, j_s, l_s}, P_{i_d, j_d, l_d}$ denote the 3D-positional encodings for positions $(i_s, j_s, l_s), (i_d, j_d, l_d)$ respectively. Note that these positional encodings *do not* contain any learnable parameters. The student $S_{img}$ performs for $N$ iterations. During *each* iteration, it performs a binding procedure and a recirculation procedure. During final evalaution step, we want to predict the features corresponding to the *last-layer L* of the VLM teacher, we set $src = (0, 0, 0)$ and iteratively set $dest = (i, j, L)$, where $1 \le i \le H, 1 \le j \le W$, generating the final $Y_{img}^{student}$. Features over spatial positions are averaged for image classification, and multiplied with the teacher's textual feature $Y_{text}$ (line 29 in algorithm 1). For image segmentation, $Y_{img}^{student}$ is directly feed-forwarded through the teacher's mask-head, where it is upsampled, and trained via standard cross-entropy loss.

### 3.2.2 THE 3D-BINDED ALGORITHM

**Binding Procedure:** Here, the student takes as input a *fixed* source $src$ location $(0, 0, 0)$ and a *random* destination $dest$ location $(i_d, j_d, l_d)$. The student should output a representation similar to teacher's representation at destination $dest$ $Y_{img}[i_d, j_d, l_d]$. We enforce this by an MSE loss:

$$L_B = \|\mathcal{S}_{img}(Src, Dest, x_{ood}) - Y_{img}[i_d, j_d, l_d]\|_2^2 \qquad \text{(line 11 in algorithm 1)} \tag{3}$$

This 'binding' procedure 'binds' the 3-D location $(i_d, j_d, l_d)$ to the teacher's output $Y_{img}(i_d, j_d, l_d)$.

**Recirculation-procedure:** Here, we sample two random locations $src1 = (i_1, j_1, l_1), src2 = (i_2, j_2, l_2)$ and a *single* destination location $dest = (i_d, j_d, l_d)$. The idea is that irrespective of whether the student operates on $(src1, dest)$ or $(src2, dest)$, it should predict the same representation everytime. We enforce this *spatial-invariance* as an additional MSE loss.

$$L_R = \|\mathcal{S}_{img}(Src1, Dest, x_{ood}) - \mathcal{S}_{img}(Src2, Dest, x_{ood})\|_2^2 \qquad \text{(line 20 in algorithm 1)} \tag{4}$$

This self-supervised step doesn't require a VLM teacher and forces the triplet $(src1, src2, dest)$ to communicate among themselves.

**Loss:** During $N$ iterations, LQN uses a combination of binding/recirculation losses. Mathematically, we supervise LQN via the loss, with $\alpha$ controlling the weight of the recirculation.

$$Loss = L_B + \alpha L_R \text{ (line 22 in algorithm 1)} \tag{5}$$

## 4 EXPERIMENTS

Next, we discuss experiments with LQN across 14 classification and 5 segmentation benchmarks.

### 4.1 TASKS AND DATASETS

Following prior works like TPT (Shu et al., 2022), we assess our LQN on two types of classification benchmarks: 1) For evaluating on natural distribution shift, we evaluate on ImageNet val (2009), along with its distribution-shifted variants, namely ImageNet-A 2021a , ImageNet-V2 (2019), ImageNet-R (2021b), and ImageNet-Sketch (2019). (2) For cross-dataset generalization tasks, we conduct experiments on 9 recognition datasets, including Flowers102 (2008), DTD (2014), OxfordPets (Parkhi et al. (2012)), UCF101 (2012), Caltech101 (2004), Food101 (2014), SUN397 (2010), FGVCAircraft (2013), and EuroSAT (2019). Additionally, we evaluate LQN on *dense* segmentation tasks. We also report results on COCO (2014) and ADE20K (2017) for panoptic, Cityscapes (2016) and ADE20K (2017) for semantic, and COCO (2014) for instance segmentation.

Table 1: **Robustness under natural distribution shifts:** Results for ImageNet and 4 distribution-shifted variants (ImageNet-A, -V2, -R, -Sketch). *Requirements* column specifies resources needed during test-time: ✗ indicates that no external data is required; *Aug. Views* denotes reliance on multiple augmented views of each test sample; *History* indicates adaptation using cumulative information from prior test samples; and *Labeled Data* denotes the use of labeled training data. LQN adapts using only a *single* test sample while achieving superior performance across distribution shifts.

| Method | Requirements | ImageNet↑ | ImageNet-A↑ | ImageNet-V2↑ | ImageNet-R↑ | ImageNet-Sketch↑ | Avg↑ | OOD Avg↑ |
|---|---|---|---|---|---|---|---|---|
| CLIP-ViT-B/16(t) | ✗ | 66.7 | 47.8 | 60.8 | 73.9 | 46.0 | 59.1 | 57.2 |
| Ensemble | ✗ | 68.3 | 49.8 | 61.8 | 77.6 | 48.2 | 61.2 | 59.4 |
| TPT[NeurIPS'22] | Augmentations | 68.9 | 54.7 | 63.4 | 77.0 | 47.9 | 62.4 | 60.8 |
| Diff-TPT[ICCV'23] | Augmentations | 70.3 | 55.6 | 65.1 | 75.0 | 46.8 | 62.5 | 60.6 |
| MTA + TPT[CVPR'24] | Augmentations | 70.0 | 58.0 | 64.2 | 78.3 | 49.6 | 64.0 | 62.5 |
| APM[NeurIPS'24] | ✗ | 68.1 | 52.1 | 67.2 | 76.5 | 49.3 | 62.6 | 61.2 |
| GS-Bias[ICML'25] | Augmentations | **70.5** | 56.6 | 64.6 | 80.4 | 50.3 | 64.5 | 63.0 |
| TPS[WACV'25] | Augmentations | 70.1 | **60.0** | 64.7 | 80.2 | 49.9 | 64.9 | 63.7 |
| LQN-[src][Ours] | ✗ | 69.4 | 54.5 | 68.3 | 78.0 | 51.0 | 64.3 | 62.7 |
| LQN [Ours] | ✗ | 70.2 | 58.6 | 68.5 | 80.4 | 50.4 | 65.6 | 64.4 |
| TDA [CVPR'24] | History | 69.5 | 60.1 | 64.6 | 80.2 | 50.5 | 64.9 | 63.8 |
| DMN-ZS[CVPR'24] | History | 72.2 | 58.2 | 65.1 | 78.5 | 53.2 | 65.4 | 63.7 |
| DPE[NeurIPS'24] | History | 71.9 | 59.6 | 65.4 | 80.4 | 52.2 | 65.9 | 64.4 |
| CoOp[IJCV'22] | Labeled Data | 71.5 | 49.7 | 64.2 | 75.2 | 47.9 | 61.7 | 59.2 |
| CoCoOp[CVPR'22] | Labeled Data | 71.0 | 50.6 | 64.0 | 76.1 | 48.7 | 62.1 | 59.9 |
| TPT + CoOp | Labeled Data | 73.6 | 57.9 | 66.8 | 77.2 | 49.2 | 64.9 | 62.8 |
| TPT + CoCoOp | Labeled Data | 71.0 | 58.4 | 64.8 | 78.6 | 48.4 | 64.3 | 62.6 |
| MTA + Coop[CVPR'24] | Labeled Data | 73.9 | 59.2 | 66.9 | 78.2 | 49.9 | 65.6 | 63.5 |
| CLIP ViT-L/14(t) | ✗ | 76.2 | 69.6 | 72.1 | 85.9 | 58.8 | 72.5 | 71.6 |
| APM[NeurIPS'24] | ✗ | 77.3 | 71.8 | 72.8 | 87.1 | 62.2 | 74.2 | 73.4 |
| LQN-[src][Ours] | ✗ | 78.6 | 74.2 | 74.3 | 89.1 | 64.1 | 76.1 | 75.3 |
| LQN[Ours] | ✗ | **78.9** | 73.7 | 75.0 | 89.4 | 63.9 | **76.2** | **75.5** |

Table 2: **Cross-dataset generalization from ImageNet to fine-grained classification tasks.** Results are reported as top-1 accuracy across nine datasets. CoOp and CoCoOp are tuned on ImageNet with 16-shot labeled data per class, whereas CLIP, ensemble prompting, TPT, APM, and our LQN require no training data or annotations.

| Method | Requirements | Flower102 | DTD | Pets | UCF101 | Caltech101 | Food101 | SUN397 | Aircraft | EuroSAT | Avg |
|---|---|---|---|---|---|---|---|---|---|---|---|
| CoOp[IJCV'22] | Labeled Data | 68.7 | 41.9 | 89.1 | 66.5 | 93.7 | 85.3 | 64.2 | 18.5 | 46.4 | 63.9 |
| CoCoOp[CVPR'22] | Labeled Data | 70.9 | 45.5 | 90.5 | 68.4 | 93.8 | 84.0 | 66.9 | 22.3 | 39.2 | 64.6 |
| TDA[CVPR'24] | History | 71.4 | 47.4 | 88.6 | 70.6 | 94.2 | 86.1 | 67.6 | 23.9 | 58.0 | 67.5 |
| DPE[NeurIPS'24] | History | 75.0 | 54.2 | 91.1 | 70.4 | 94.8 | 86.1 | 70.0 | 28.9 | 55.7 | 69.4 |
| CLIP-ViT-B/16(t) | ✗ | 67.4 | 44.3 | **88.3** | 65.1 | 93.4 | 83.7 | 62.6 | 23.7 | 42.0 | 63.6 |
| Ensemble | ✗ | 67.0 | 45.0 | 86.9 | 65.2 | 93.6 | 82.9 | 65.6 | 23.2 | 50.4 | 64.6 |
| TPT[NeurIPS'22] | Augmentations | 69.0 | 47.8 | 87.8 | 68.0 | **94.2** | 84.7 | 65.5 | 24.8 | 42.4 | 65.1 |
| DiffTPT[ICCV'23] | Augmentations | **70.1** | 47.0 | 88.2 | 62.6 | 92.4 | **87.2** | 65.7 | 25.6 | 43.1 | 65.4 |
| MTA[CVPR'24] | Augmentations | 68.0 | 45.9 | 88.2 | 68.6 | 94.2 | 85.0 | 66.6 | 25.2 | 45.3 | 65.2 |
| APM[NeurIPS'24] | ✗ | 62.0 | 48.9 | 81.6 | 72.6 | 89.6 | 84.2 | 65.7 | 29.7 | 55.7 | 65.5 |
| GS-Bias[ICML'25] | Augmentations | 71.9 | 46.1 | 90.3 | 67.5 | 94.6 | 86.0 | 67.4 | 26.4 | 52.4 | 67.0 |
| LQN-[src][Ours] | ✗ | 65.2 | 50.0 | 84.1 | 72.2 | 93.8 | 85.6 | 67.0 | 29.9 | 56.1 | 67.0 |
| LQN[Ours] | ✗ | 66.8 | **51.3** | 85.0 | **73.1** | 94.0 | 86.4 | 67.6 | 30.5 | 57.0 | **67.9** |

## 4.2 IMPLEMENTATION DETAILS

We implement two variants of our model: *LQN-[src]* and *LQN*. *LQN-[src]* means that the model only uses a *single location*, i.e. $dest$ and *does not* perform any recirculation-procedure. *LQN* is the variant which uses *both* binding/recirculation procedure. These variants are optimized with Adam using a learning rate of $1 \times 10^{-4}$. Input images are normalized to ImageNet statistics. The total number of iterations are set to $T = 15$. For LQN, we set the recirculation loss weight to $\alpha = 0.7$. The GFLOPs are measured using Meta's *fvcore* package (FLOPs $\times 10^9$). To ensure statistical reliability, we report the mean accuracy of three runs with different seeds.

While performing recirculation in the LQN model, there are $n = H \times W \times L$ plausible locations. Selecting a triplet $(src_1, src_2, dest)$ yields $\binom{n}{3} = O(n^3)$ possibilities, making exhaustive computation infeasible. However, we find that randomly sampling as few as $5\%$ of these triplets provides sufficient performance gains, consistent with Masked Autoencoders He et al. (2022), where most tokens can be dropped without loss in performance.

## 4.3 MAIN RESULTS

**Baselines** We compare with (1) zero-shot VLMs such as CLIP, including backbones of varying sizes like ViT-B/16 and ViT-L/14; (2) relevant TTT baselines such as TPT and Diff-TPT, which adapt VLMs using augmented views of a *single* test sample, and TPS, which provides an efficient alternative by learning shift vectors for each class prototype; and (3) various TTA baselines such as TDA, DMN-ZS, and DPE, which rely on historical information from *multiple* test samples, (4)

Table 3: Comparison across various segmentation tasks. GFLOPs consumed by Panoptic and Instance segmentation are identical. Both use an input resolution of $1280^2$, whereas semantic segmentation uses a resolution of $1024^2$. PQ: Panoptic Quality, AP: Average Precision.

| Method | Backbone | GFLOPs ↓ | Panoptic | | Instance | Semantic | | | |
|---|---|---|---|---|---|---|---|---|---|
| | | | COCO | ADE20K | COCO | CityScapes | | ADE20K | |
| | | | PQ ↑ | PQ ↑ | AP ↑ | GFLOPs ↓ | mIoU ↑ | GFLOPs ↓ | mIoU ↑ |
| Mask2Former [CVPR 2022] | ViT-Adapter-L | 4817 | 59.7 | 53.0 | 51.4 | 5200 | 84.5 | 910 | 58.9 |
| EoMT(t) [CVPR 2025] | ViT-L | 4146 | 58.3 | 51.7 | 48.8 | 4350 | 84.2 | 721 | 58.4 |
| APM [NeurIPS 2024] | MLP | 4336 | 59.2 | 52.6 | 51.6 | 4540 | 85.1 | 911 | 58.5 |
| LQN-[src] [Ours] | MLP | 4342 | 59.9 | 53.2 | 52.1 | 4490 | 85.7 | 861 | 61.2 |
| LQN [Ours] | MLP | 4384 | **61.8** | **55.4** | **53.8** | 4588 | **86.3** | 959 | **62.7** |

as well as prompt learning approaches like CoOp and CoCoOp, which require annotated training data and often incur additional computational or memory overhead. In contrast, LQN is designed specifically for the TTT setting: it adapts the VLM using only *one* test sample without requiring auxiliary training data or multiple test streams, while also leveraging an ensemble of 80 prompts similar to CLIP to improve robustness.

**Natural Distribution Shifts:** In Tab 1, we compare the performance of our LQN on in-domain ImageNet and its 4 OOD (Out-Of-Distribution) variants. Zero-Shot CLIP underperforms in the OOD case, obtaining a mere 57.2 average accuracy. Compared with other TTT-baselines, LQN on average outperforms TPT by 3.2%, TPS by 0.7%, Diff-TPT by 3.1%, and ZERO by 1.6%. On the ImageNet-val set, LQN comes close 0.1% to Diff-TPT. However, note that Diff-TPT requires a heavily parameterized diffusion model to generate test augmentations, whereas LQN is lightweight with $25M$ parameters. Methods like CoOP and CocoOp utilize annotated training data, which limits their effectiveness in real-world situations. This might pose problems in scenarios where models are 'rolled out' on edge-devices, and *only* test samples are available. Despite this, our method still exhibits significant performance gains of 5.2% compared to CoOp. Leveraging CLIP VIT-L/14 as a teacher, LQN outperforms the teacher model by 3.9%.

**Cross-Dataset Generalization:** In Tab 2, we evaluate our LQN on 9 additional fine-grained recognition benchmarks. LQN-[src] obtains an SOTA average accuracy of 67.0, which is 1.5% better than the prior SOTA method APM. LQN improves the performance further to 67.9, and notably outperforms even strong TTA baselines like TDA/DPE. On 4/9 datasets, we come close to other methods, and acknowledge the potential for further improvements.

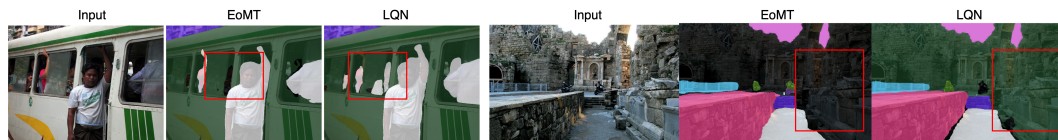

Figure 3: LQN demonstrates better qualitative segmentation results than EoMT teacher. (left) LQN can even segment persons 'occluded' behind the window of the bus. (right) LQN semantically-groups visual-elements of the scene, including the walls, whereas EoMT falls short.

**Generalization to Segmentation Tasks:** Building on the strong performance of LQN in classification, we extend our study to the more challenging dense-level segmentation setting. Table 3 reports results with segmentation methods, including the Mask2Former baseline Cheng et al. (2022) and the recent EoMT architecture Kerssies et al. (2025). Although EoMT substantially reduces FLOPs, it performs worse than Mask2Former in terms of accuracy. By using EoMT as the teacher, our LQN attains superior performance relative to both baselines. Importantly, even with TTT, the FLOPs consumed by LQN remain lower than those of the Mask2Former baseline. Fig3 shows some sample qualitative segmentation results of LQN.

## 4.4 Analysis

**Computation FLOP Analysis:** We analyze how LQN achieves faster convergence compared to popular TTT methods such as TPT, TPS, and DiffTPT, which rely on augmenting each test sample multiple times. In terms of the full *test-time tuning* process, as shown in Fig. 4(ii), TPT requires only one gradient update per test sample but constructs 63 augmented views, resulting in $63 + 1 = 64$ forward passes through the encoder and a total cost of 1312 GFLOPs. By contrast, LQN performs multiple iterations. The first forward step costs 20.5 GFLOPs through CLIP's encoder, while the subsequent 14 backward steps cost 10 GFLOPs each for updating the student, yielding a total of

| Method | H | Time | Acc. | Gain |
|--------|---|------|------|------|
| CLIP ResNet-50 | ✗ | 9 min | 59.8 | - |
| TDA | ✓ | 16m | 61.3 | +1.5 |
| DPE | ✓ | 1h 50m | 63.4 | +3.6 |
| TPT | ✗ | 9h 15m | 60.7 | +0.93 |
| DiffTPT | ✗ | >20 h | 60.8 | +0.99 |
| TPS | ✗ | 55 min | 61.4 | +1.6 |
| APM | ✗ | 1h 7m | 61.6 | +1.8 |
| LQN-[src] | ✗ | 47m | 61.9 | +2.1 |
| LQN | ✗ | 1h 27m | 62.3 | +2.5 |

Figure 4: **(Left)** Wall-clock time comparison on ImageNet using CLIP ResNet-50. H: ✓ means TTT used cumulative training results from multiple history test samples. **(Right)** Graphs of computation cost and performance. (i) Test-time inference computation cost. (ii) Test-time tuning computation cost. (iii) Semantic segmentation performance.

$20.5 + 14 \times 10 = 160.5$ GFLOPs—an order of magnitude lower than TPT. As shown in Tab. 1, LQN not only converges faster but also surpasses TPT in accuracy. Moreover, unlike TPT, which incurs additional FLOPs for optimizing prompts in the text encoder, LQN avoids this overhead.

Fig. 4(i) further highlights the *test-time inference* cost. To obtain *last-layer* features, TPT requires feed-forward through all 12 layers of CLIP's encoder (20.5 GFLOPs), whereas LQN can query its student $S$ directly at a constant cost of 10 GFLOPs. Finally, Tab. 4 reports the *actual wall-clock time* Zhang et al. (2024) required for TTT over all 50,000 ImageNet validation samples on a single A6000 GPU (following DPE). TPS, the prior SOTA, consumes 55m since it avoids backpropagation through the text encoder. LQN-[src] further reduces this to 47m while improving accuracy by 0.5%. Full LQN achieves even higher accuracy (62.3%), though with an added cost of 1h 27m due to recirculation. Thus, while LQN improves over both TPT and TPS, it introduces a trade-off between accuracy and wall-clock efficiency.

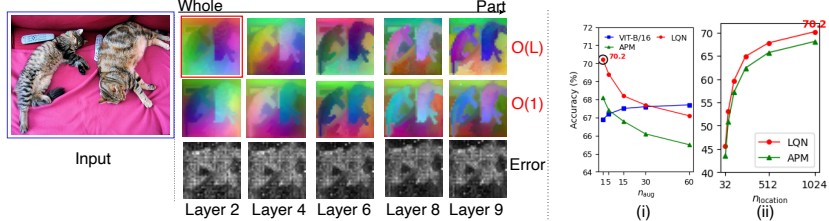

Figure 5: **(Left)** Token visualizations of the teacher model EoMT (top row), the student model LQN (middle row), and their difference (bottom row). EoMT operates sequentially, predicting intermediate features in $O(L)$ time, whereas LQN predicts them in constant $O(1)$ time. As depth increases, LQN's predictions closely match those of EoMT. **(Right) (i)** Effect of the number of image augmentations, comparing ViT-B/16, APM, and LQN. Augmenting the test sample during TTT *harms* APM and LQN. **(ii)** Effect of the number of randomly sampled token positions. Distilling *more* teacher locations into APM/LQN improves performance, highlighting the benefit of single-sample supervision.

**LQN *generalizes* to unseen teacher layers beyond those used in test-time training:** In Fig. 4(iii), we use EoMT ViT-L as LQN's teacher for semantic segmentation on ADE20k. When TTT is performed by distilling *all* 24 layers of the teacher into LQN, we obtain a peak mIoU of 62.7, thereby improving over the EoMT baseline of 58.4 mIoU. Interestingly, we next perform TTT using only the first 20 layers of EoMT. During inference, however, we query LQN with the deeper layers 21–24. In this case, we still observe an increase in performance, reaching 60.3 mIoU. This result shows that LQN has effectively encoded depth as a valid spatial dimension, enabling it to generalize to the teacher's deeper layers *never seen* during training.

**LQN can predict teacher's features in *constant* time *irrespective* of layer-depth:** In Fig 5 (Left), we feed-forward a sample image from the COCO dataset into both the EoMT teacher and LQN. We plot intermediate representations via t-SNE reduction. We observe that the predicted features are significantly similar to those of the teacher. A notable advantage of LQN is that estimating those features takes $O(1)$ time, whereas in EoMT it takes $O(l)$, where $l$ is the depth of the queried layer.

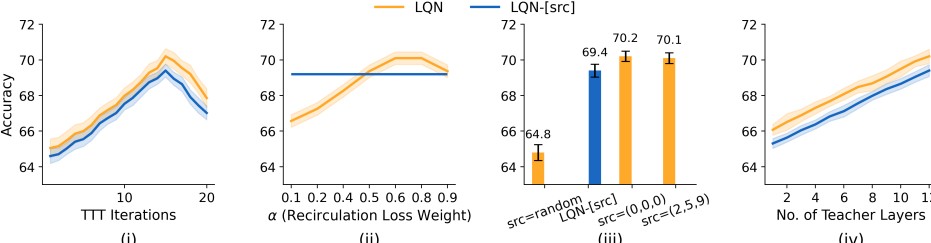

Figure 6: Ablations on LQN-[src] and LQN variants: (i) TTT iteration $N$ study. (ii) Effect of recirculation loss coefficient $\alpha$. (iii) Effect of different 'Src' locations in LQN. (iv) Effect of the number of teacher layers during TTT.

## 4.5 ABLATIONS STUDIES

Recall, we implemented two variants of our model, i) LQN-[src], which only contains *dest*, and *does not* use the recirculation procedure. ii) LQN which uses *both* binding and recirculation procedures. We ablate several important hyperparameters in this section. To ensure consistency, we perform all TTT experiments on the ImageNet val set, with CLIP VIT-B/16 as LQN's teacher.

**Effect of varying TTT iterations** $N$**:** In Fig. 6(i), we study the impact of increasing the number of iterations for LQN. We observe that performance first improves, peaks at 15 iterations, and then begins to decrease. It is important to note that, before adapting *each* test sample, LQN's weights are initialized *randomly* from a normal distribution. In contrast, methods such as TPT and TPS adapt the pre-trained VLM itself, which has been trained on large-scale data. This fundamental difference likely explains why LQN requires multiple iterations ($N > 1$) to achieve optimal adaptation.

**Effect of the recirculation loss weight** $\alpha$**:** In Fig. 6(ii), we observe that performance improves as $\alpha$ increases and peaks at $\alpha = 0.7$. This result is notable because a higher weight on the recirculation objective means the network benefits more from the self-supervised consistency constraint rather than relying solely on the teacher's supervision. It highlights how LQN's predicted features can eventually *surpass* those of the frozen VLM image encoder. This suggests that recirculation acts as a strong regularizer, encouraging more robust and transferable representations.

**Modeling 'pairs' of teacher tokens is important:** LQN operates on pairs of locations $(src, dest)$. A natural question is whether such pairing is even *necessary*. As shown in Fig. 6(iii), LQN-[src], which only uses a single location *dest*, achieves $69.4\%$, whereas modeling both $(src, dest)$ in LQN improves performance to $70.2\%$. Similarly, during test-time inference, we set $src = (0, 0, 0)$. Why is this fixed choice needed? We observe that $src = random$ drops the performance significantly to $64.8\%$. Interestingly, fixing $src$ to another constant position, e.g., $(2, 5, 8)$, achieves performance comparable to $(0, 0, 0)$. This suggests that $src$ should be a consistent fixed 3D location when decoding features for the last layer, but its exact choice is not critical.

**Increasing teacher layers distilled into LQN improves performance:** In Fig. 6(iv), we observe that distilling a larger number of teacher layers into LQN consistently improves performance. This suggests that modeling intermediate teacher layers enables the student to capture multi-level feature representations, which directly correlates with stronger downstream TTT performance, highlighting the benefit of leveraging hierarchical depth from the teacher.

**LQN *can* adapt using just a *single* test sample:** How can adaptation occur even without *test-time augmentation*? We study this by subjecting the student to multiple augmented views of the same test sample. In Fig. 5(i), we find that increasing augmentations *improves* the ViT-B based teacher, consistent with prior observations in TPT and TPS. However, we observe *lower* results in LQN and APM, which points to a *unique inductive bias* in their structure. Recall that the LQN student takes coordinate-based inputs $(src, dest)$. For a test sample $x_{ood}$, we vary the number of such coordinate-based locations distilled into LQN during TTT. As shown in Fig. 5(ii), distilling *more teacher locations* further improves performance. A formal justification of this unique property of coordinate-conditioned networks has been discussed in Hinton (2022; 2023); Modi & Rawat (2024a).

$L_2$ **outperforms entropy minimization:** Applying entropy-minimization objective (similar to TPT) on LQN *drops* performance from $70.2$ to $67.4$, implying $L_2$ constraint on features is more effective.

## 5    CONCLUSION

Our results suggest that it is possible to adapt VLM's like CLIP using *as few as* 1 test-sample (TTT), and *without* requiring primitives like data augmentation. Further, we can design architectures like LQN which can predict intermediate features of a teacher in a *constant* amount of time, as opposed to $O(l)$ time which is incurred in *classical* neural-nets Bengio et al. (2006). This brings us closer towards validating the insight that perception is a 'continuous' field which can be 'queried' (Layer *Query* Network), instead of computed 'sequentially' Hinton (2023); Löwe et al. (2019). We remain encouraged by LQN's potential to improve asynchronous processing across different layers.

## 6    REPRODUCIBILITY STATEMENT

To facilitate reproducibility, we have included the detailed implementation details and hyperparameters in the supplementary material. The complete codebase and model checkpoints will be made publicly available following the review process. LQN is designed to be lightweight and can be trained on a single GPU (e.g., NVIDIA Pascal). For larger-scale runs, it also supports parallelization across a multi-node setup—for instance, a cluster with 2 nodes, each equipped with 8 NVIDIA A6000 (Ampere) GPUs. We provide more details relevant to reproducibility in supplementary (Sec 8).

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

# 7 BROADER IMPACT

There are two core ideas that motivate the design of the Layer Query Network (LQN). The first idea is that positional encodings could function as an addressing mechanism Vaswani et al. (2017). When a network is conditioned on a specific positional encoding, it becomes capable of 'retrieving' the relevant entity stored at that location Modi & Rawat (2024b). The second idea is to recognize that feature-representations in a neural network form part-whole hierarchiesAmir et al. (2021): they can be treated as a 'graph-like structure', where each node represents a part/whole. Directed relationships between such nodes $(src, dest)$ (denoted by $\leftarrow, \rightarrow$) could then be modeled using the LQN architecture. This allows mapping part-whole hierarchies onto a shared connectionist hardwareHinton (2023). However, given a single node, one *cannot* infer all its incoming/outgoing edges *without brute-forcing* over all the possible entity pairs and identifying the ones where the network's response becomes high.

**Limitations:** In future, we would like to study how such co-ordinate based nets perform when trained on large-scale data. In its current formulation, LQN relies on recirculation, which is a compute-intensive procedure. We are excited to explore how this time could be reduced further. Ultimately, we would like to extend LQN to videos: *mimicking* how spatio-temporal processing occurs in infero-temporal pathway of human primates.

**Ethical Considerations:** Layer Query Network (LQN) introduces a non-sequential mechanism for querying intermediate representations in deep models, offering computational and adaptability benefits. However, this capability raises important ethical considerations. At test time, the model may adapt in ways that unintentionally amplify biases present in the teacher or the test distribution, particularly in safety-critical settings. Furthermore, the ability to synthesize features at arbitrary depths may reduce interpretability and complicate efforts to audit or explain model behavior. While LQN reduces inference-time overhead, it builds upon large pre-trained teacher models with significant resource footprints. We emphasize that LQN should not be deployed in applications involving surveillance, manipulation, or deceptive content generation, and advocate for its use in settings that prioritize transparency, accountability, and societal benefit.

# 8 IMPLEMENTATION DETAILS

**Architecture:** In Tab 4, we inflate the full architecture of our LQN. LQN consists of a single CNN filter. Given an input image $x_{ood}$, we first run a single convolution filter on it with a stride $s$. The resultant $h/s \times w/s \times 1$ vector is then 'flattened'. The two locations $(src, dest)$ are encoded as 3-D positional encodings, along with the flag positional encoding (Alg 1). The flattened input image, along with $d$ dimensional source, and $d$ dimensional destination are passed through an MLP. The first layer of the MLP contains $(h/s * w/s + d_p + d_p^\dagger) * 4096$ learnable parameters. The subsequent layers contain 4096 * 4096, 4096 * 4096, 4096 * 2048 , 2048 * 1024 parameters respectively. Finally, we have a projection head which projects the MLP output to $1024, d$, where $d$ is the dimensionality of internal tokens of a teacher.

Table 4: **LQN architecture for TTT**: with input dimensions $h, w, c$ and feature dimension $d_p$: dimensionality of positional encoding. $s$: stride of convolutional filter in encoder, $d_c$: dimension of the intermediate token of teacher on which LQN learns. LQN contains two locations, $(src, dest)$, so additional $d_p$ term is added to the input. $\dagger$ : LQN-[src] contains only a single location, so this term is not added.

|  | Layer | Feature Dimension (H × W × C) | $n_{kernels}$ | Stride | Padding Input / Output |
|---|---|---|---|---|---|
|  | Input | $h \times w \times c$ |  |  |  |
| Encoder | Conv | $h/s \times w/s \times d$ | 1 | $s$ | 0 / 0 |
| Decoder | Linear | $(h/s * w/s + d_p + d_p^\dagger) * 4096$ | - | - | - |
|  | Linear | 4096 * 4096 | - | - | - |
|  | Linear | 4096 * 4096 | - | - | - |
|  | Linear | 4096 * 2048 | - | - | - |
|  | Linear | 2048 * 1024 | - | - | - |
| Feature Projection Head | Linear | $1024 * d_c$ | - | - | - |

**Hyperparameters:** All hyper-parameters utilized for LQN during test-time-training are detailed in 5. We leveraged the seed 0/7/42 in most of our experiments. The weight matrices in LQN were initilized with from a random distribution with $\mu = 0$ and $\sigma = 0.01$. We tried initializing the net with other schemes such as xavier initialization etc, but found normal-weight intialization to work the best. All our code has been written in Pytorch version 1.13.0. However, since we only use pytorch, and no external libraries, we expect that our codebase will also support more recent versions, for eg, Pytorch 2.0+. We also note that performing test-time-training with 16 bit floating point allows us to effectively use recent GPU architectures for eg, Ampere: they contain a larger number of tensor cores *in addition* to CUDA cores which results in significant speedups during the exprimentation process. Finally, we normalize an input image using standard Imagenet stats, and *dont resort to any other form of augmentation*, thereby making the pipeline far-simpler.

Table 5: **LQN hyperparameters** during test-time-training.

| | |
|---|---|
| Number of Test samples | 50000 (Imagenet Splits), variable for other datasets. |
| Testing iterations | 15 |
| Batch Size | 1 |
| Learning Rate | 1e-4 |
| Optimizer | Adam |
| Feature Output size $d$ | 768/1024 |
| Positional Encoding size | 768/1024 |
| Image/Crop Size | 448 |
| Augmentations | Normalization, $\mu = (0.485, 0.456, 0.406)$, $\sigma = (0.229, 0.224, 0.225)$ |
| Precision | fp16 (grad-scaled) |
| Num of Workers | 8 |
| Operating System | 1x rtx a6000 48GB/96GB ram/Ubuntu 22.04/2TB ssd/5TB HDD |

**ViT Encoder:** During our experiments in test-time-training, APM relies on higher-dimensional intermediate token distilled from a teacher trained on a large-scale-dataset, often via contrastive image-text objectives. We showed quantitative results with CLIP VIT-L/VIT-H and semantic clusterings with EoMT (CVPR'25).

CLIP is a zero-shot model from OpenAI which contains a vision encoder, and a textual encoder. The textual encoder tokenises input class names to features. Both image/text encoder project them to common dimensionality, and classification happens by measuring distances in contrastive space, thereby offering a higher degree of freedom, as opposed to training a class-sensitive linear-probe. CLIP VIT-L features an output CLS token of 768 dimensions, while CLIP VIT-H outputs 1024 dimensions both of which have been accommodated in Tab4.

## 9 DETAILS OF THE DATASETS

Evaluating a model's robustness to distribution shifts necessitates testing on datasets that feature a broad spectrum of perturbations—such as *fog, snow, rain*, and other real-world variations. A common strategy is to apply synthetic corruptions to well-established test sets (e.g., ImageNet) to create benchmark splits suitable for controlled evaluation. Alternatively, new test sets may be manually compiled from online sources to capture modality changes, such as *sketches* or *artistic reinterpretations*. Below, we describe the key datasets used in this work to assess the robustness and generalization capabilities of LQN. These datasets consist of both classification and segmentation benchmarks.

### 9.1 CORRUPTION & DISTRIBUTION SHIFT BENCHMARKS

**CIFAR-10-C:** Comprising 10,000 test samples from CIFAR-10, this benchmark introduces 15 corruption types (e.g., blur, noise, weather effects), each applied at 5 severity levels. Our evaluation focuses on the most difficult setting—level 5—due to computational constraints.

**ImageNet-C:** A widely used benchmark for corruption robustness based on the original ImageNet dataset. It includes 15 distortion types applied at 5 severity levels, affecting all 1,000 classes. The corruptions degrade image quality in ways that simulate real-world noise and artifacts.

**ImageNet-V2:** A re-collection of ImageNet-like samples from the web, preserving label distribution across 1,000 categories. The dataset consists of 10,000 images across three distinct splits, offering a testbed for evaluating generalization to naturally shifted data.

**ImageNet-A:** Contains 7,500 images from 200 categories that are hard for standard models like ResNet-50. These examples are "naturally adversarial"—real-world images that consistently cause misclassification.

**ImageNet-R:** Focuses on stylized renditions of ImageNet classes, including paintings, cartoons, and other artistic formats. It contains approximately 30,000 images across 200 categories and evaluates robustness to stylistic domain shifts.

**ImageNet-Sketch:** A challenging modality shift dataset containing 50,000 black-and-white sketches corresponding to 1,000 ImageNet categories. It tests the model's ability to recognize abstract shapes and contours, without relying on color or texture cues.

To ensure consistency with prior work (e.g., CLIP), we evaluate LQN using an ensemble of 80 handcrafted textual prompts across these ImageNet-derived benchmarks.

### 9.2 Dense Prediction Benchmarks

**COCO (Common Objects in Context):** A large-scale dataset for object detection, segmentation, and captioning tasks. It features over 200,000 labeled images containing instances of 80 object categories in diverse, cluttered scenes. COCO is widely used to evaluate models' ability to handle multi-object, real-world environments.

**Cityscapes:** Focused on urban street scenes, this dataset contains 5,000 finely annotated images from 50 different European cities. It includes 19 semantic classes relevant to autonomous driving (e.g., road, pedestrian, traffic light) and is primarily used for evaluating semantic segmentation under real-world conditions.

**ADE20K:** A challenging benchmark for semantic segmentation that includes 25,000 images annotated with over 150 object and stuff categories. The dataset spans indoor, outdoor, urban, and natural scenes, providing a diverse set of environments for evaluating dense prediction tasks.

## 10 Additional Ablations

**How should the locations $(src1, src2, dest)$ be chosen for the recirculation procedure?** Images consist of large redundant information, for eg, a large number of pixels generally represent the same object, for eg, a tree. If $(src1, src2, dest)$ correspond to the *same* object, the LQN might not learn useful information. So, it would help if $(src1, src2, dest)$ correspond to different objects. To validate this, we run a segmentation model like EomT on an input test sample prior to beginning the TTT. We sample $(src1, src2, dest)$ from locations corresponding to different segments, and observe that the performance of the LQN model on ImageNet val using CLIP VIT-B/16 as a teacher improves from the reported 70.2 to 71.5, thereby even outperforming the Diff-TPT in Tab 1) of the main paper. This validates the insight that $(src1, src2, dest)$ should come from different objects.

However, this improvement comes at a cost: one needs to run a segmentation model before beginning to TTT, which increases the GFlops. Therefore, being able to discover object regions, *without* segmentation models remains an interesting direction of future research.

**Why do we concatenate embeddings and not add them as in a transformer:** Recall that first MLP layer of LQN-[src] contains as input the test sample's representation $x'$, and positional encoding corresponding to $(src)$. Instead of concatenating these inputs, we perform an additional experiment where positional encoding is added to x'. Specifically, given $x_{ood}$, we first convolve it with a CNN filter to yield $x' \in \mathbb{R}^{h' \times w' \times 1}$. The resultant $x'$ is flattened and added to $p_{src}$, which similar to transformer. We find that the TTT performance on ImageNet for LQN-[src] model using CLIP VIT-B/16 as a teacher *drops* from 69.4 to 61.2. However, we do note that in the original VIT paper, Dosovitskiy et al. (2021), the best results were estimated by adding the positional encoding to the image features. We believe that the reason behind this might be the unique inductive bias in the architecture of LQN, which is originally inspired by the APM paperModi & Rawat (2024b).

## 11 Pseudo-code of APM

---

**Algorithm 2 A**synchronous **P**erception **M**achine. Student operations shown in blue.

---

**Input**: Teacher Image/Text Encoder $\mathcal{T}_{img}$ / $\mathcal{T}_{text}$, Student $\mathcal{S}_{img}$ (to adapt), $N$ iterations, $P \in \mathbb{R}^{(H \times W) \times D}$ (positional embedding)

**Require:** OOD image $x_{ood}$, Class Label $cls$ & Predict class logit

1: // Teacher image and text feats
2: $Y_{img} \leftarrow \mathcal{T}_{img}(x_{ood}) \in \mathbb{R}^{H \times W \times D}$
3: $Y_{text} \leftarrow \mathcal{T}_{text}(cls)$
4: $\mathcal{S}_{img} \leftarrow 0$                                                      // Reset the student
5: **for** iteration $k \in N$ **do**
6:     $i, j \leftarrow$ Sample(H,W)                                       // random spatial index
7:     $P_{i,j} \leftarrow P(i, j)$                                         // Position embedding
8:     loss $\leftarrow \|\mathcal{S}_{img}(P_{i,j}, x_{ood}) - Y_{img}[i, j]\|_2^2$
9:     Update $\mathcal{S}_{img}$                                             // update student
10: **end for**
11: //Spatial average of $x_{ood}$ via $S_{img}$
12: **for** $\forall (i, j) \in (H, W)$ **do**
13:     $P_{i,j} \leftarrow P(i, j)$
14:     $Y_{img}^{Student}$ += $\mathcal{S}_{img}(P_{i,j}, x_{ood}) / (H \cdot W)$
15: **end for**
16: Teacher Text + Student image produces logit
17: $P_{cls} \leftarrow Y_{img}^{Student} \cdot Y_{text}$
18: **Output:** $P_{cls}$

---

**Test-Time Training (TTT) setup:** Following recent work, APM (Modi & Rawat, 2024a) (shown in algorithm 2), we adopt a teacher student distillation framework (fig. 1(b)). The teacher is a frozen VLM (*e.g.* CLIP, ViT-L), while the student $S$ is *optimized* at test time for $N$ iterations. First the teacher produces text and images embedding corresponding to $x_{ood}$ and class label $cls$, *i.e.* $Y_{img}$, and $Y_{text}$ (eq. (1)). For $N$ iterations, the students randomly select a batch of spatial indices (i,j) and tries to *mimic* teacher embeddings on those spatial indices. At every iteration, the student $S$ is reset before doing any prediction (ideal TTT), and uses positional embedding (Vaswani et al., 2017) and the input image $x_{ood}$ to generate image features. MSE loss is used to train the student. After $N$ iterations, the student uses the teacher text encoder to generate the class logit. It has been shown to improve light-weight students to inherit (& surpass) teacher's zero-shot generalization on OOD samples.

## 12 ADDITIONAL RESULTS ON CIFAR 10-C

We report more results on Cifar 10C in Table 6, where LQN model gets the lowest error rate of 13.0.

Table 6: **CIFAR-10-C** results at *highest* severity level of 5. We report Error Rate (%, lower is better). The t- model acts as the teacher for APM and LQN variants. TTT was performed on the test set with randomly initialized weights. APM and LQN weights were reinitialized after each TTT iteration to prevent information leakage. All LQN variants outperform prior methods, with LQN-Two Word achieving the best overall performance.

| Method | orig | gauss | shot | impul | defoc | glass | motn | zoom | snow | frost | fog | brit | contr | elas | pixel | jpeg | Avg |
|---|---|---|---|---|---|---|---|---|---|---|---|---|---|---|---|---|---|
| TTT-Online | 8.2 | 25.8 | 22.6 | 30.6 | 14.6 | 34.4 | 18.3 | 17.1 | 20.0 | 18.0 | 16.9 | 11.2 | 15.6 | 21.6 | 18.1 | 21.2 | 19.1 |
| UDA-SS | 9.0 | 28.2 | 26.5 | 20.8 | 15.6 | 43.7 | 24.5 | 23.8 | 25.0 | 24.9 | 17.2 | 12.7 | 11.6 | 22.1 | 20.3 | 22.6 | 21.4 |
| Zeroshot | | | | | | | | | | | | | | | | | |
| CLIP ViT-L/14 | 4.63 | 35.4 | 32.3 | 21.9 | 19.3 | 49.7 | 19.3 | 17.3 | 17.0 | 15.1 | 21.6 | 8.4 | 15.9 | 34.6 | 25.0 | 27.4 | 24.5 |
| CLIP ViT-L/14 (t) | | | | | | | | | | | | | | | | | |
| APM | 3.5 | 21.9 | 30.1 | 13.7 | 15.2 | 34.1 | 11.9 | 11.1 | 15.0 | 9.0 | 13.5 | 5.8 | 9.5 | 23.0 | 15.8 | 17.0 | 14.8 |
| LQN-[src] (Ours) | 3.2 | 20.1 | 21.3 | 13.2 | 14.7 | 31.5 | 11.1 | 10.8 | 14.3 | 8.7 | 12.4 | 5.2 | 8.8 | 20.5 | 14.5 | 15.5 | 13.9 |
| LQN (Ours) | 2.9 | 18.8 | 20.0 | 12.5 | 14.1 | 30.2 | 10.5 | 10.1 | 13.4 | 8.1 | 11.7 | 4.9 | 8.1 | 19.3 | 13.6 | 14.3 | 13.0 |

## 13 ADDITIONAL RESULTS ON IMAGENET-C

Following TTT-MAEGandelsman et al. (2022), we evaluate our method , we evaluate our method on ImageNet-C. ImageNet-C is a dataset which consists of 15 types of corruptions applied to the original ImageNet validation set. As evidenced in Tab 7, we obtain the highest accuracy of 53.0 on the highest severity level 5, thereby showcasing the efficacy of LQN.

Table 7: **LQN's performance on ImageNet-C, level 5**. The first three rows are fixed models without test-time training. The third row, ViT probing, is the baseline used in Gandelsman et al. (2022). A ✓ in P means that method leveraged **pre-trained weights** on clean variant of train set aka, Image-net and downstream-ttt on corrupted version. OpenCLIP VIT-L/14 is generally more robust. LQN does better on various noises with an average accuracy score of 52.3.

| | P | brigh | cont | defoc | elast | fog | frost | gauss | glass | impul | jpeg | motn | pixel | shot | snow | zoom | Average |
|---|---|---|---|---|---|---|---|---|---|---|---|---|---|---|---|---|---|
| Joint Train | ✓ | 62.3 | 4.5 | 26.7 | 39.9 | 25.7 | 30.0 | 5.8 | 16.3 | 5.8 | 45.3 | 30.9 | 45.9 | 7.1 | 25.1 | 31.8 | 24.8 |
| Fine-Tune | ✓ | 67.5 | 7.8 | 33.9 | 32.4 | 36.4 | 38.2 | 22.0 | 15.7 | 23.9 | 51.2 | 37.4 | 51.9 | 23.7 | 37.6 | 37.1 | 33.7 |
| ViT Probe | ✓ | 68.3 | 6.4 | 24.2 | 31.6 | 38.6 | 38.4 | 17.4 | 18.4 | 18.2 | 51.2 | 32.2 | 49.7 | 18.2 | 35.9 | 32.2 | 29.2 |
| TTT-MAE | ✓ | 69.1 | 9.8 | 34.4 | 50.7 | 44.7 | 50.7 | 30.5 | 36.9 | 32.4 | 63.0 | 41.9 | 63.0 | 33.0 | 42.8 | 45.9 | 44.4 |
| OpenCLIP VIT-L/14(t) | ✗ | 71.9 | 47.0 | 50.3 | 32.7 | 58.3 | 46.9 | 26.0 | 26.5 | 28.1 | 62.7 | 37.7 | 58.3 | 28.2 | 50.4 | 37.9 | 42.1 |
| APM | ✗ | 77.4 | 51.9 | 56.6 | 37.9 | 64.8 | 53.2 | 28.7 | 31.4 | 33.0 | 68.4 | 44.1 | 64.5 | 33.1 | 56.9 | 43.9 | 50.3 |
| LQN-[src] (Ours) | ✗ | 79.2 | 54.0 | 58.9 | 40.2 | 67.1 | 55.3 | 30.8 | 33.5 | 35.2 | 70.8 | 46.3 | 66.7 | 35.1 | 59.1 | 46.0 | 51.8 |
| LQN (Ours) | ✗ | **80.1** | **55.2** | **60.4** | **41.5** | **68.7** | **56.4** | **31.6** | **34.2** | **36.0** | **71.9** | **47.2** | **67.9** | **36.3** | **60.3** | **47.1** | **53.0** |

Table 8: **Performance on ImageNet-C, level 4**. The first two rows are from the supplementary materials of Gandelsman et al. (2022). A ✓ in column P indicates use of **pre-trained weights** on clean ImageNet followed by TTT on corrupted inputs. OpenCLIP ViT-L/14 shows stronger robustness than earlier models. LQN variants surpass prior methods.

| | P | brigh | cont | defoc | elast | fog | frost | gauss | glass | impul | jpeg | motn | pixel | shot | snow | zoom | Average |
|---|---|---|---|---|---|---|---|---|---|---|---|---|---|---|---|---|---|
| Baseline | ✓ | 73.1 | 33.1 | 35.8 | 56.9 | 54.2 | 45.2 | 39.6 | 26.0 | 38.2 | 62.0 | 43.2 | 60.3 | 32.2 | 44.2 | 40.7 | 47.4 |
| TTT-MAE | ✓ | 72.7 | 39.6 | 45.7 | 64.9 | 58.3 | 52.6 | 48.5 | 42.8 | 47.6 | 67.0 | 50.5 | 66.6 | 42.4 | 45.7 | 51.5 | 53.2 |
| OpenCLIP VIT-L/14 | ✗ | 74.2 | 64.2 | 58.7 | 57.8 | 66.3 | 52.8 | 45.3 | 34.6 | 45.2 | 68.9 | 46.6 | 63.9 | 41.1 | 56.2 | 45.6 | 54.8 |
| APM | ✗ | 79.2 | 70.4 | 64.9 | 63.7 | 72.3 | 58.6 | 51.2 | 40.4 | 51.3 | 74.1 | 53.0 | 70.0 | 46.7 | 62.5 | 51.8 | 59.6 |
| LQN-[src] (Ours) | ✗ | 80.1 | 71.2 | 65.7 | 64.4 | 73.1 | 59.2 | 52.0 | 41.1 | 52.1 | 74.7 | 53.6 | 70.7 | 47.3 | 63.2 | 52.6 | 60.4 |
| LQN (Ours) | ✗ | **80.9** | **72.1** | **66.5** | **65.2** | **73.9** | **59.9** | **52.9** | **41.9** | **52.8** | **75.3** | **54.2** | **71.4** | **47.9** | **63.9** | **53.3** | **61.2** |

Table 9: **Performance on ImageNet-C, level 3**. The first two rows are from the supplementary materials of Gandelsman et al. (2022). A ✓ in column P indicates that the method used **pre-trained weights** on clean ImageNet and applied TTT on the corrupted set. OpenCLIP ViT-L/14 is more robust than earlier models. Both LQN variants outperform prior approaches.

| | P | brigh | cont | defoc | elast | fog | frost | gauss | glass | impul | jpeg | motn | pixel | shot | snow | zoom | Average |
|---|---|---|---|---|---|---|---|---|---|---|---|---|---|---|---|---|---|
| Baseline | ✓ | 75.8 | 62.7 | 49.5 | 67.1 | 59.8 | 47.6 | 57.1 | 35.0 | 57.4 | 68.6 | 60.2 | 70.1 | 54.3 | 54.7 | 48.0 | 57.6 |
| TTT-MAE | ✓ | 75.8 | 64.4 | 59.4 | 71.2 | 64.0 | 54.0 | 63.6 | 50.7 | 64.2 | 71.3 | 64.2 | 73.1 | 61.8 | 58.0 | 57.4 | 64.4 |
| OpenCLIP VIT-L/14 | ✗ | 75.8 | 71.8 | 65.5 | 67.7 | 69.0 | 54.7 | 58.9 | 42.4 | 59.5 | 72.8 | 59.9 | 69.7 | 58.2 | 63.5 | 51.8 | 62.5 |
| APM | ✗ | 80.5 | 77.2 | 71.3 | 73.3 | 74.8 | 60.6 | 64.7 | 48.5 | 65.4 | 77.8 | 61.6 | 75.2 | 64.1 | 69.3 | 58.0 | 68.5 |
| LQN-[src] (Ours) | ✗ | 81.2 | 78.0 | 72.0 | 74.0 | 75.4 | 61.2 | 65.5 | 49.2 | 66.1 | 78.4 | 62.1 | 75.9 | 64.8 | 69.9 | 58.8 | 69.2 |
| LQN (Ours) | ✗ | **81.9** | **78.8** | **72.8** | **74.9** | **76.2** | **61.9** | **66.3** | **49.9** | **66.9** | **79.1** | **62.8** | **76.7** | **65.5** | **70.5** | **59.5** | **69.9** |

Table 10: **Performance on ImageNet-C, level 2**. The first two rows are from the supplementary materials of Gandelsman et al. (2022). A ✓ in column P indicates that the method used **pre-trained weights** on clean ImageNet and performed TTT on the corrupted version. OpenCLIP ViT-L/14 is generally more robust than earlier models. Both LQN variants outperform prior methods.

| | P | brigh | cont | defoc | elast | fog | frost | gauss | glass | impul | jpeg | motn | pixel | shot | snow | zoom | Average |
|---|---|---|---|---|---|---|---|---|---|---|---|---|---|---|---|---|---|
| Baseline | ✓ | 77.4 | 71.2 | 62.3 | 51.0 | 66.3 | 58.4 | 68.6 | 59.2 | 64.9 | 70.4 | 70.6 | 74.7 | 66.2 | 54.2 | 55.2 | 64.1 |
| TTT-MAE | ✓ | 77.8 | 71.5 | 69.4 | 49.7 | 69.8 | 62.7 | 72.5 | 66.4 | 70.0 | 72.7 | 72.3 | 76.2 | 70.6 | 58.7 | 63.6 | 68.3 |
| OpenCLIP VIT-L/14 | ✗ | 76.6 | 74.4 | 71.4 | 53.8 | 72.0 | 62.6 | 67.6 | 64.0 | 64.6 | 73.8 | 69.0 | 72.8 | 66.4 | 61.8 | 58.3 | 66.1 |
| APM | ✗ | 81.1 | 79.4 | 76.6 | 59.4 | 77.3 | 68.2 | 73.1 | 70.0 | 70.3 | 78.6 | 74.5 | 77.8 | 72.0 | 67.8 | 64.3 | 72.4 |
| LQN-[src] (Ours) | ✗ | 81.7 | 80.2 | 77.5 | 60.1 | 78.1 | 69.0 | 73.9 | 70.6 | 71.0 | 79.3 | 75.1 | 78.5 | 72.6 | 68.4 | 65.1 | 73.2 |
| LQN (Ours) | ✗ | **82.4** | **81.0** | **78.4** | **60.9** | **78.9** | **69.8** | **74.7** | **71.3** | **71.7** | **80.1** | **75.8** | **79.2** | **73.4** | **69.1** | **65.8** | **74.1** |

Table 11: **APM and LQN performance on ImageNet-C, level 1**. The first two rows are reproduced from the supplementary materials of Gandelsman et al. (2022). A ✓ in column P indicates that the method used **pre-trained weights** on clean ImageNet and applied TTT on the corrupted set. Open-CLIP VIT-L/14 is generally more robust than earlier models. Both LQN variants surpass OpenCLIP and prior methods.

| | P | brigh | cont | defoc | elast | fog | frost | gauss | glass | impul | jpeg | motn | pixel | shot | snow | zoom | Average |
|---|---|---|---|---|---|---|---|---|---|---|---|---|---|---|---|---|---|
| Baseline | ✓ | 78.5 | 74.5 | 68.1 | 73.9 | 70.5 | 70.6 | 74.8 | 68.6 | 72.3 | 73.0 | 75.2 | 75.9 | 73.6 | 69.3 | 63.7 | 71.4 |
| TTT-MAE | ✓ | 78.9 | 74.7 | 72.5 | 74.7 | 72.9 | 72.2 | 76.8 | 72.9 | 75.5 | 74.5 | 75.8 | 77.0 | 75.9 | 71.9 | 69.3 | 73.1 |
| OpenCLIP VIT-L/14 | ✗ | 77.3 | 75.4 | 73.5 | 73.1 | 73.5 | 71.4 | 71.9 | 70.2 | 69.9 | 75.1 | 73.7 | 74.2 | 71.9 | 71.2 | 65.2 | 71.1 |
| APM | ✗ | 81.6 | 80.3 | 78.6 | 78.0 | 78.6 | 76.6 | 77.2 | 75.7 | 75.1 | 79.6 | 78.7 | 79.1 | 76.9 | 76.4 | 70.7 | 76.0 |
| LQN-[src] (Ours) | ✗ | 82.4 | 81.1 | 79.4 | 78.8 | 79.2 | 77.3 | 77.9 | 76.2 | 75.9 | 80.1 | 79.3 | 79.8 | 77.5 | 77.1 | 71.6 | 76.8 |
| LQN (Ours) | ✗ | **83.2** | **82.0** | **80.3** | **79.6** | **80.1** | **78.0** | **78.6** | **77.0** | **76.5** | **80.9** | **80.2** | **80.5** | **78.4** | **77.9** | **72.3** | **77.6** |

## 14 ADDITIONAL QUALITATIVE RESULTS

In the original paper, we showed LQN's results on segmentation. In Fig7,8,10, we qualitatively demonstrate the semantic quality of the predictions made by our model. We compare against the

EoMT baseline, APM, and our method. As it can be seen, LQN consistently gives better outputs than other methods.

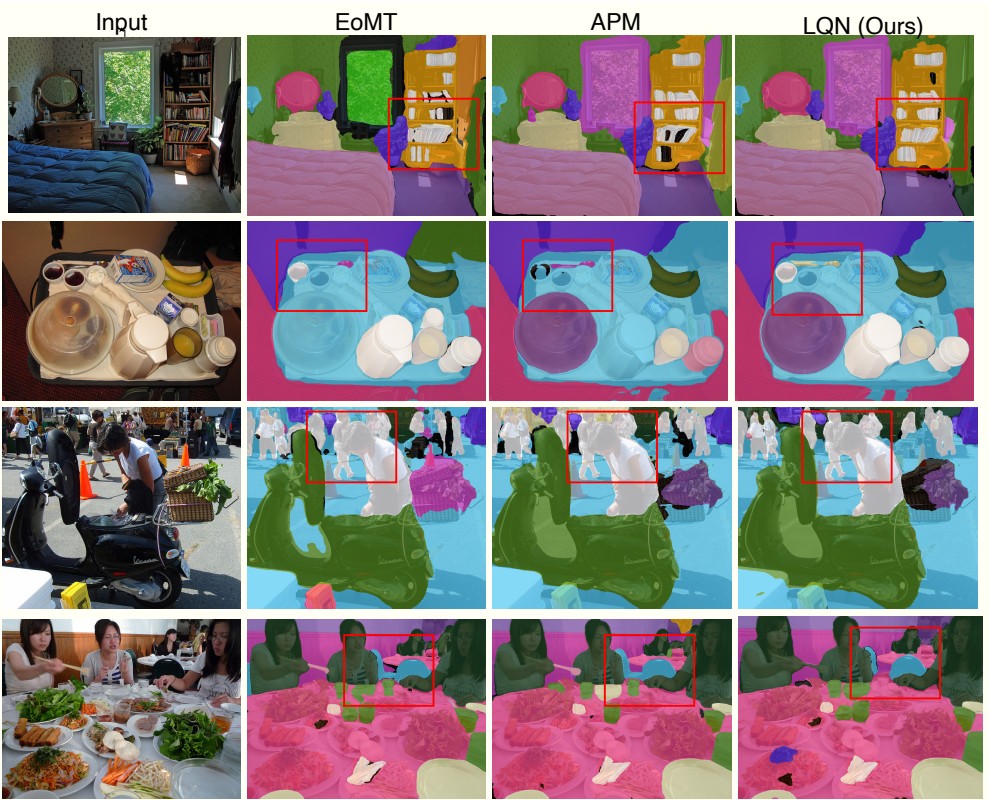

Figure 7: **Qualitative results on LQN.** Panoptic segmentation on COCO-Val set. LQN obtains more semantically-detailed masks than the EoMT/APM baselines via test-time-training. Masks visualized after 15 iterations of TTT on both APM/LQN. EoMT is a fully-supervised, *fixed* baseline. TTT on APM/LQN is performed with EoMT as the teacher. The red regions are the regions where a kind reader can focus on to see the comparison in the prediction quality. Starting from above, LQN easily segments the books in the bookshelf as white region, spoon as a white outline on the plate, distinguishes between the people in the background, and easily partitions chairs into two distinct parts, whereas other EoMT and APM group them together. The red regions are the regions where a kind reader can focus on to see the comparison in the prediction quality. Starting from above, LQN easily segments the books in the bookshelf as white region, spoon as a white outline on the plate, distinguishes between the people in the background, and easily partitions chairs into two distinct parts, whereas other EoMT and APM group them together.

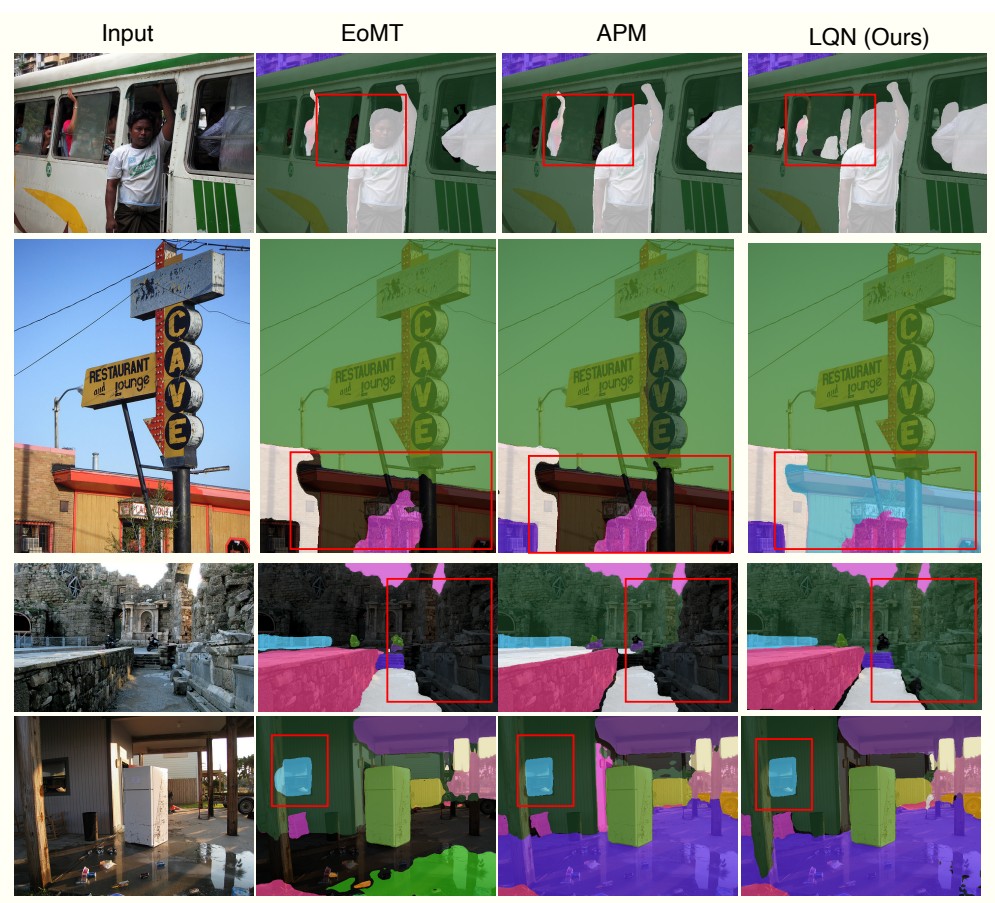

Figure 8: **Qualitative results on LQN.** Panoptic segmentation on COCO-Val set. LQN obtains more semantically-detailed masks than the EoMT/APM baselines via test-time-training. Masks visualized after 15 iterations of TTT on both APM/LQN. EoMT is a fully-supervised, *fixed* baseline. TTT on APM/LQN is performed with EoMT as the teacher. The red regions are the regions where a kind reader can focus on to see the comparison in the prediction quality. Starting from above, (first row) LQN gets the passengers in the bus in the background, even though they are partially occluded, (second row) easily segments the building into the blue regions, (third row) partitions buildings into a green colored region whereas other two methods dont detect it as all/term it as background (fourth row) gets window in the proper shape.

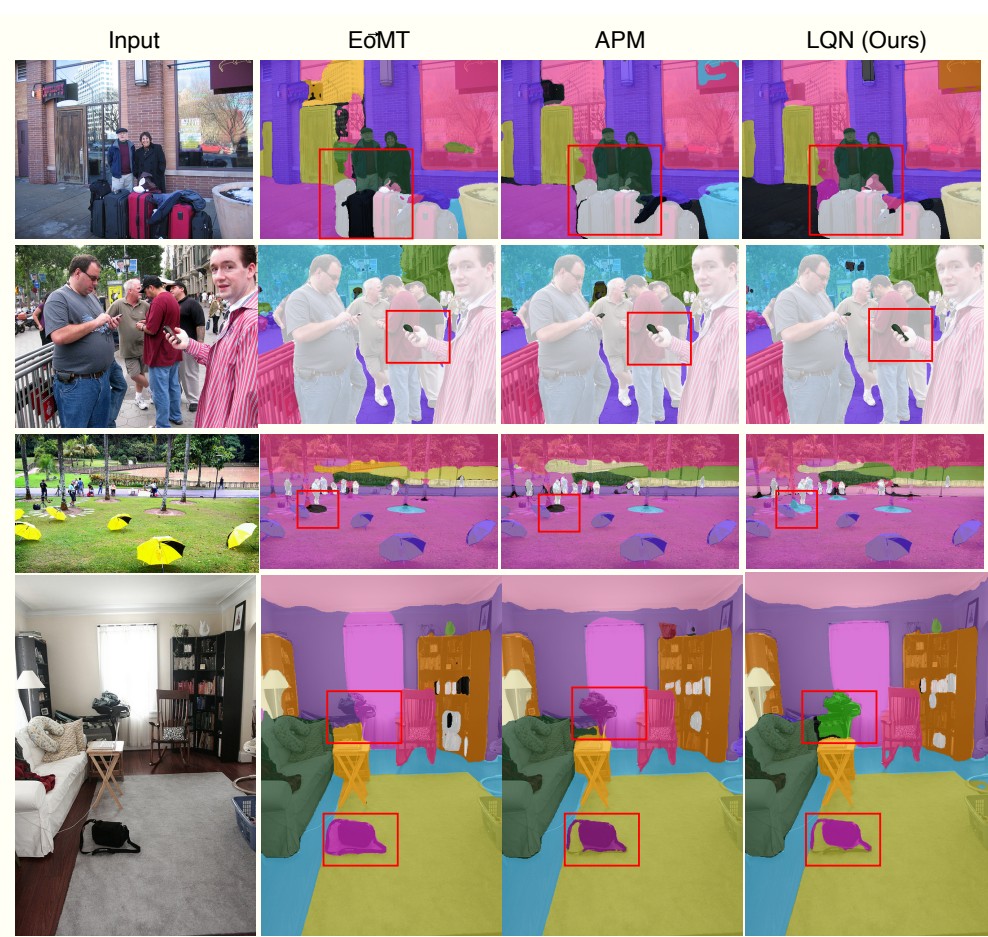

Figure 9: **Qualitative results on LQN.** Panoptic segmentation on COCO-Val set. LQN obtains more semantically-detailed masks than the EoMT/APM baselines via test-time-training. Masks visualized after 15 iterations of TTT on both APM/LQN. EoMT is a fully-supervised, *fixed* baseline. TTT on APM/LQN is performed with EoMT as the teacher. The red regions are the regions where a kind reader can focus on to see the comparison in the prediction quality. Starting from above, (first row) LQN properly segments the suitcases, (second row) Precise boundaries of the cellphone in the person's hand, even though part of the cellphone is gripped very tightly by the person's hand. (third row) segments the base of the tree, whereas other two methods dont detect it at all (fourth row) is able to understadn the fine regions which correspond to the bag/floor. In contrast, EoMT/APM confuse that some part of the purse is also the background.

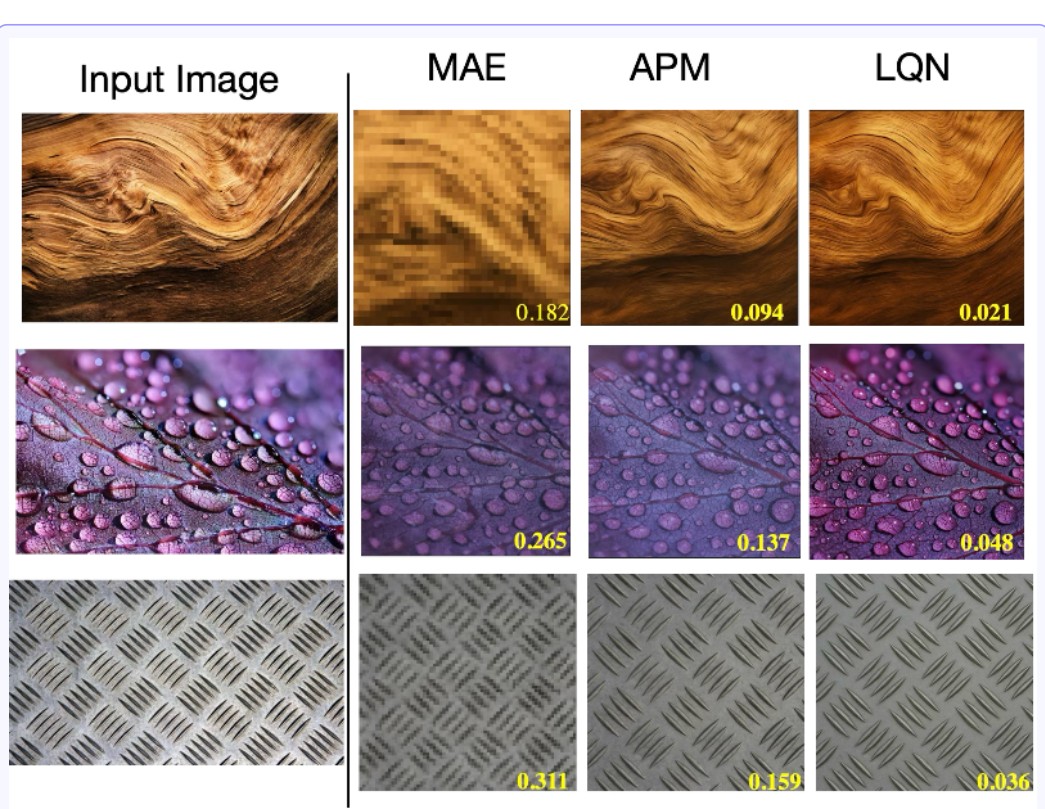

Figure 10: **Qualitative results on LQN.** LQN reconstructs RGB images with *lowest* pixelwise loss as compared to MAE/APM. This illustrates the extreme scenario when input images contain repeating textures and tests whether the model encodes semantics consistently.

