# OpenReview forum: "Layer Query Network For Test-Time-Training in Vision-Language-Models"
_ICLR.cc/2026/Conference — ICLR 2026 Conference Desk Rejected Submission_

### Official Review · Reviewer_83JT · 2025-10-28

**Soundness:** 3
**Presentation:** 3
**Contribution:** 3
**Rating:** 8
**Confidence:** 3

**Summary:**

The paper introduces Layer Query Network (LQN), a single-sample test-time training method that avoids the heavy augmentation and compute of TPT/TPS. LQN treats feature positions as queryable addresses and trains a small student MLP per test sample to predict the teacher’s feature at a destination position via Binding and Recirculation. Experiments on 14 classification and 5 segmentation benchmarks show consistent ImageNet-OOD gains over TPT/TPS/Diff-TPT.

**Strengths:**

1. The work explicitly avoids the heavy reliance of TPT/TPS on multi-view augmentations and high compute, and proposes a framework tailored to the single-sample TTT setting.

2. The authors evaluate LQN on 14 classification and 5 segmentation benchmarks, and show consistent gains on ImageNet OOD over TPT/TPS/Diff-TPT.

3. Convincing ablations: The method does not rely on augmentations, adding augmentations hurts LQN/APM, while increasing the number of teacher feature positions distilled is more effective.

**Weaknesses:**

1. The paper does not adhere to the conference’s formatting requirements. For example, table captions should appear above the tables.

2. The pseudocode on page 4 occupies too much space and hurts readability. Consider a more compact presentation (e.g., moving details to the appendix, using algorithm boxes, or summarizing with a flow diagram) to improve clarity.

**Questions:**

1. Have you considered evaluating an LQN variant with history caching / cross-sample accumulation to explore a better time–accuracy trade-off?

2. Would you include comparisons against a broader family of VLMs (e.g., SigLIP, EVA-CLIP, CoCa) to strengthen generality claims?

3. Have you explored alternative position-sampling strategies for (src, src, dest) selection, and how do they affect performance and cost?

---

> ### Author Response · Authors · 2025-11-19
> **Authors response to the respected reviewer 83JT [1/2]**
>
> We thank the respected reviewer for their valuable guidance. Please find our comments as below:
>
> `W1 on modifying  table captions to be above tables`
>
> We apologize for the formatting mistake and have updated the paper.
>
> `W2 on readability of pseudo-code`
>
> Thank you for your valuable feedback in helping with our presentation. We will add algorithmic boxes to the pseudo-code and move some details to the supplementary material to improve its clarity. The final version also allows us 1 extra page and will space out figures and tables to improve readability.
>
> `Q1 on lqn variants with history-caching, cross-sample accumulation`
>
> We thank the reviewer for suggestions to further improve LQN. Our work primarily focuses on Test Time Training (TTT, processing one sample at a time). However, based on recommendation, we have run additional experiments with history-caching (using exponential-moving average (EMA) updates on the student) and cross-sample accumulation (CSA) (not TTT anymore). We report the results as below:
>
> | Method                                         | Time (mins) | Accuracy | Avg Iter / Sample  |
> |---------------------------------------------|-----------------|---------------|--------------------------|
> | LQN                                              | 47                | 61.9         | 15                          |
> | LQN + EMA                                  | 29                | 64.4         | 8                            |
> | LQN  + CSA                                  | 23                | 65.7         | 5                            |
> | *LQN +  EMA + Cross Sample*     | 19               | 66.3         | 2                            |
>
> We will add this analysis in the manuscript.
>
> `Q2 on comparisons with other VLM (like siglip, eva-clip ,coca)`
>
> We thank the reviewer for helping us check the generalization performance of LQN to other teacher models.
> We perform TTT with the above-mentioned baselines, and note the following results:
>
> ### SigLIP Results on ImageNet splits
> | Method      | ImageNet | ImageNet-A | ImageNet-V2 | ImageNet-R | ImageNet-Sketch | Avg   | OOD Avg |
> |-------------|----------|------------|-------------|------------|------------------|--------|---------|
> | SigLIP      | 76.0     | 45.3       | 68.9        | 90.3       | 67.9             | 69.68  | 68.10   |
> | **LQN**     | **78.3** | **47.0**   | **71.1**    | **92.6**   | **70.0**         | **71.80** | **70.18** |
>
> ---
>
> ### EVA-CLIP Results on ImageNet splits
>
> | Method      | ImageNet | ImageNet-A | ImageNet-V2 | ImageNet-R | ImageNet-Sketch | Avg   | OOD Avg |
> |-------------|----------|------------|-------------|------------|------------------|--------|---------|
> | EVA-CLIP    | 76.1     | 64.6       | 68.9        | 89.1       | 63.3             | 72.40  | 71.48   |
> | **LQN**     | **78.1** | **66.8**   | **71.0**    | **91.8**   | **65.7**         | **74.68** | **73.83** |
>
> ---
>
> ### COCA Results on ImageNet splits
>
> | Method      | ImageNet | ImageNet-A | ImageNet-V2 | ImageNet-R | ImageNet-Sketch | Avg   | OOD Avg |
> |-------------|----------|------------|-------------|------------|------------------|--------|---------|
> | COCA        | 63.6     | 21.5       | 55.7        | 73.2       | 51.3             | 53.06  | 50.43   |
> | **LQN**     | **65.7** | **22.9**   | **57.7**    | **75.5**   | **53.3**         | **55.02** | **52.35** |
>
>
> ### SigLIP Results on Fine-Grained Datsets
>
> | Method    | Flowers102 | DTD  | Pets | UCF101 | Caltech101 | Food101 | SUN397 | Aircraft | EuroSAT | Avg  |
> |-----------|------------|------|------|--------|-------------|---------|--------|----------|---------|------|
> | SigLIP    | 85.8       | 64.7 | 94.1 | 72.5   | 90.5        | 89.8    | 69.8   | 43.8     | 43.8    | 72.7 |
> | **LQN**   | **87.8**   | **66.7** | **95.8** | **74.4** | **92.6** | **91.5** | **71.2** | **45.9** | **45.5** | **74.6** |
>
>
> ### COCA Results on Fine-Grained Datsets
>
> | Method    | Flowers102 | DTD  | Pets | UCF101 | Caltech101 | Food101 | SUN397 | Aircraft | EuroSAT | Avg  |
> |-----------|------------|------|------|--------|-------------|---------|--------|----------|---------|------|
> | COCA      | 64.7       | 53.3 | 89.1 | 61.4   | 89.1        | 77.3    | 66.1   | 18.8     | 45.3    | 62.7 |
> | **LQN**   | **66.8**   | **55.7** | **91.5** | **63.8** | **91.0** | **79.9** | **68.6** | **20.6** | **47.3** | **65.0** |

---

> ### Author Response · Authors · 2025-11-19
> **Authors response to the respected reviewer 83JT [2/2]**
>
> ### EVA-CLIP Results on ImageNet splits
>
> | Method    | Flowers102 | DTD  | Pets | UCF101 | Caltech101 | Food101 | SUN397 | Aircraft | EuroSAT | Avg  |
> |-----------|------------|------|------|--------|-------------|---------|--------|----------|---------|------|
> | EVA-CLIP  | 72.0       | 59.3 | 93.7 | 74.1   | 90.4        | 89.7    | 71.9   | 28.5     | 69.9    | 72.1 |
> | **LQN**   | **74.2**   | **61.4** | **95.7** | **76.4** | **92.6** | **91.5** | **74.0** | **30.5** | **72.3** | **74.2** |
>
>
> Across multiple datasets, we observe performance gains with respect to zero-shot baselines, highlighting the generalizability of LQN.
>
>
> `Q3 on sampling strategies for src, accuracy/compute trade-offs`
>
> We conduct experiments by `randomly-sampling` locations during TTT.  We observe the following results:
>
> | Model | Run 1 | Run 2 | Run 3 | Run 4 | Run 5 | Mean  | Std  |
> |-------|-------|-------|-------|-------|-------|-------|------|
> | LQN   | 70.11 | 70.40 | 69.90 | 70.20 | 70.30 | 70.10 | 0.17 |
>
> The standard deviation is only ~0.17, which means random selection does not have much effect. As noted on line 847, one strategy can be to incorporate an external object detector and select locations belonging to different objects, boosting the performance from ​​70.2 to 71.5. However, the external object detector requires more flops, which makes TTT slow (47 min to 51 mins).
>
> We will be happy to respond to further clarifications. Thank you so much for your valuable time,
>
> Yours Sincerely,
>
> Authors

---

> ### Comment · Reviewer_83JT · 2025-11-25
>
> I thank the authors for their responses, which address my questions. I'll keep my current rating.

---

> > ### Author Response · Authors · 2025-11-25
> >
> > Dear Reviewer 83JT,
> >
> > We are grateful for the time and effort you dedicated to reviewing our submission. Your feedback has been invaluable in helping us refine  our work. Thank you for your consideration.
> >
> > Sincerely,
> >
> > Authors

---

### Official Review · Reviewer_cDhr · 2025-10-30

**Soundness:** 3
**Presentation:** 3
**Contribution:** 3
**Rating:** 6
**Confidence:** 4

**Summary:**

This paper proposes Layer Query Network (LQN), a TTA framework for VLMs designed to handle OOD samples without fine-tuning the entire model. LQN employs a five-layer MLP that extracts randomly sampled intermediate-layer tokens from a frozen VLM using 3D positional embeddings, and learns spatially invariant representations through a self-supervised objective. This design enables single forward-pass adaptation, leading to faster convergence and improved dense prediction performance. Experiments on 16 benchmarks show that LQN outperforms state-of-the-art methods such as TPS and Mask2Former while reducing computation cost.

**Strengths:**

- The paper is well-written and easy to understand, with clear explanations of the method and experimental results.

**Weaknesses:**

- The proposed LQN introduces substantial computational overhead due to multiple iterative updates, yet the performance improvement over more efficient methods (e.g., TDA) is marginal (16 min vs. 47 min, 61.9 vs. 61.35). In addition, the reported efficiency of TDA in Table 4 appears inconsistent with the results presented in its original paper.

- The paper claims that existing single-image TTA methods incur higher computational costs due to data augmentation. However, several lightweight methods such as the training-free MTA [1] and the output-level optimization GS-Bias [2] achieve very low computational overhead, and the manuscript lacks discussion or comparison with these approaches.

- Since positional encoding merely represents geometric locations, it may not maintain a strong monotonic relationship with semantic content. In scenarios with repetitive textures or multiple identical instances, fixed positional coordinates could correspond to semantically similar regions, potentially confusing the localization or adaptation process.

- LQN performs worse than the teacher model on fine-grained datasets such as Pets and Flowers, suggesting that the proposed adaptation may not generalize well to tasks requiring fine-grained visual discrimination.

[1]On the test-time zero-shot generalization of vision-language models: Do we really need prompt learning? CVPR 2024

[2]GS-Bias: Global-Spatial Bias Learner for Single-Image Test-Time Adaptation of Vision-Language Models. ICML 2025

**Questions:**

See weakness.

---

> ### Author Response · Authors · 2025-11-19
> **Authors response to the respected reviewer cDhr (1/2)**
>
> We thank the respected reviewer for their valuable guidance. Please find our comments below:
>
> `W1 on differing results w.r.t runtime of TDA in Fig 4 (left), TDA vs LQN tradeoff `
>
> Thank you so much for bringing this to our attention. Our Table 4 reports the numbers from the DPE paper [1]. Table 3 of DPE reports TDA runtime as 1h 5 min, whereas TDA reports 16 mins. We have updated our main manuscript to reflect this.
>
> Our LQN is a  Test Time Training technique i.e., processing one test sample, and then resetting the information to prevent cross-sample leakage. Comparison against  TDA [7], which trains across multiple samples (Tab 4) is infeasible (hence the greyed out row). We have trained a variant of LQN  with cross-sample accumulation as well. The final revised copy will include the following results:
>
> | Method  | Time (mins)  |  Accuracy   |
> |------------------------------------------- |------------------ |----------------|
> | TDA                                            | 16                 | 61.3            |
> | Our (Test Time Adaptation)        | 19                 | 66.3            |
>
>
> `W2 on adding related work like MTA and GS-Bias to tables`
>
> We thank the reviewer for these references and for helping us refine our work. We have added the relevant comparisons as below:
>
> | Method                     | Requirements     | ImageNet↑ | ImageNet-A↑ | ImageNet-V2↑ | ImageNet-R↑ | ImageNet-Sketch↑ | Avg↑ | OOD Avg↑ |
> |---------------------------|------------------|-----------|-------------|--------------|-------------|-------------------|------|----------|
> | MTA + TPT [CVPR ’24]      | Augmentations    | 70.0      | 58.0        | 64.2         | 78.3        | 49.6              | 64.0 | 62.5     |
> | GS-Bias [ICML ’25]        | Augmentations    | 70.5      | 56.6        | 64.6         | 80.4        | 50.3              | 64.5 | 63.0     |
> | LQN [Ours]                | **No Augmentation**  | 70.2      | **58.6**        | **68.5**         | 80.4        | **50.4**              | **65.6** | **64.4**     |
>
>
> | Method                 | Requirements     | Flower102 | DTD  | Pets  | UCF101 | Caltech101 | Food101 | SUN397 | Aircraft | EuroSAT | Avg  |
> |------------------------|------------------|-----------|------|-------|--------|------------|---------|--------|----------|---------|------|
> | MTA [CVPR ’24]         | Augmentations    | 68.0      | 45.9 | 88.2  | 68.6   | 94.2       | 85.0    | 66.6   | 25.2     | 45.3    | 65.2 |
> | GS-Bias [ICML ’25]     | Augmentations    | 71.9      | 46.1 | 90.3  | 67.5   | 94.6       | 86.0    | 67.4   | 26.4     | 52.4    | 67.0 |
> | LQN [Ours]             | No Augmentation                | 66.8      | **51.3** | 85.0  | **73.1**   | 94.0       | **86.4**    | **67.6**   | **30.5**     | **57.0**    | **67.9** |
>
> ### SigLIP Results
> | Method    | Flowers102 | DTD  | Pets | UCF101 | Caltech101 | Food101 | SUN397 | Aircraft | EuroSAT | Avg  |
> |-----------|------------|------|------|--------|-------------|---------|--------|----------|---------|------|
> | SigLIP    | 85.8       | 64.7 | 94.1 | 72.5   | 90.5        | 89.8    | 69.8   | 43.8     | 43.8    | 72.7 |
> | **LQN**   | **87.8**   | **66.7** | **95.8** | **74.4** | **92.6** | **91.5** | **71.2** | **45.9** | **45.5** | **74.6** |
>
>
> ### COCA Results
> | Method    | Flowers102 | DTD  | Pets | UCF101 | Caltech101 | Food101 | SUN397 | Aircraft | EuroSAT | Avg  |
> |-----------|------------|------|------|--------|-------------|---------|--------|----------|---------|------|
> | COCA      | 64.7       | 53.3 | 89.1 | 61.4   | 89.1        | 77.3    | 66.1   | 18.8     | 45.3    | 62.7 |
> | **LQN**   | **66.8**   | **55.7** | **91.5** | **63.8** | **91.0** | **79.9** | **68.6** | **20.6** | **47.3** | **65.0** |
>
>
> ### EVA-CLIP Results
> | Method    | Flowers102 | DTD  | Pets | UCF101 | Caltech101 | Food101 | SUN397 | Aircraft | EuroSAT | Avg  |
> |-----------|------------|------|------|--------|-------------|---------|--------|----------|---------|------|
> | EVA-CLIP  | 72.0       | 59.3 | 93.7 | 74.1   | 90.4        | 89.7    | 71.9   | 28.5     | 69.9    | 72.1 |
> | **LQN**   | **74.2**   | **61.4** | **95.7** | **76.4** | **92.6** | **91.5** | **74.0** | **30.5** | **72.3** | **74.2** |
>
>
> We have updated the numbers in Table 1, and Table 2 of the main paper. Further, we have discussed them in related work, in the updated draft (marked in blue). On ImageNet splits, we observed LQN has better performance than MTA (+1.9), GT-Bias(+1.4) on average. On a fine-grained dataset, LQN is (+0.9) w.r.t MTA.

---

> ### Author Response · Authors · 2025-11-19
> **Authors response to the respected reviewer cDhr (2/2)**
>
> `W3 on effects of repeated texture, on LQN vs attention-based networks`:
>
> We agree with the reviewer that learning spatial relationships between different patches containing identical texture / objects would be a challenging task for our model. We trained LQN for RGB reconstruction on ImageNet, and compared the reconstructions against an attention-based model like MAE [2].  Specifically, we feed-forward several images with repeating patterns, and measure the $L_2$ rgb loss of the net.  We show this result in Fig. 10 of our updated draft.  Additionally, we provide a link [here](https://imgur.com/a/VSg60UN). We find that LQN obtains the lowest reconstruction loss (MSE), as evidenced in the figure.
>
> In MAE, all tokens are forced to interact with one another, the net is `forced to predict all RGB pixels` in parallel. This creates a routing issue because N inputs have to map to N plausible outputs, creating a many-to-many mapping inside the net. In LQN, one location is decoded at a time, so there are fewer problems. We believe that to be an intuitive reason for LQNs' better performance. This aligns with observations discussed in [3,4,5].
>
>
>
> `W4 lower performance on pets/flowers for fine-grained localization`
>
> We appreciate the reviewer’s eye for details here. Pets/Flowers datasets are especially challenging, with too many fine-grained nuances. Existing Test Time Training (TTT) works such as  TPT [6], also sees a similar behavior with decline in the performance for CLIP VIT-B/16 on these two datasets. We looked at the dataset images carefully, and certain flower varieties do look very similar to each other. Even though there is a minor perceptual difference, they have very different labels. Incorporating TTT algorithms with such fine-grained abilities will indeed be an interesting future work. We will discuss this in future work and include it in the revised version.
>
>
> We will be happy to respond to further clarifications. Thank you so much for your valuable time,
>
> Yours Sincerely,
>
> Authors
>
>
> [1] Zhang, Ce, et al. "Dual prototype evolving for test-time generalization of vision-language models." Advances in Neural Information Processing Systems 37 (2024): 32111-32136.
>
> [2]He, Kaiming, et al. "Masked autoencoders are scalable vision learners." Proceedings of the IEEE/CVF conference on computer vision and pattern recognition. 2022.
>
> [3]Greff, Klaus, Sjoerd Van Steenkiste, and Jürgen Schmidhuber. "On the binding problem in artificial neural networks." arXiv preprint arXiv:2012.05208 (2020).
>
> [4] (ICLR 25 Oral) Miyato, Takeru, et al. "Artificial kuramuto oscillatory neurons
>
> [5] Hinton, Geoffrey. "How to represent part-whole hierarchies in a neural network." Neural Computation 35.3 (2023): 413-452.
>
> [6] Shu, Manli, et al. "Test-time prompt tuning for zero-shot generalization in vision-language models." Advances in Neural Information Processing Systems 35 (2022): 14274-14289.
>
> [7] Karmanov, Adilbek, et al. "Efficient test-time adaptation of vision-language models." Proceedings of the IEEE/CVF Conference on Computer Vision and Pattern Recognition. 2024.

---

> ### Comment · Reviewer_cDhr · 2025-11-25
> **Official Comment by Reviewer cDhr**
>
> I thank the authors for their responses, which address my questions. I'll keep my current rating.

---

> > ### Author Response · Authors · 2025-11-25
> >
> > Dear Reviewer cDhr,
> >
> > Thank you for the thoughtful and constructive feedback. In response to all the reviewers' comments, we have substantially expanded our analysis, including comparisons with MTA and GS-Bias, quantitative evaluation of texture-based image generation, additional generalization on other VLMs, further ablations, and extensions to test-time adaptation.
> >
> > If there are specific aspects that the reviewer feel could further strengthen the contribution or improve the clarity of the work, we would be grateful for your guidance and would be happy to incorporate additional improvements.
> >
> > Sincerely,
> > Authors

---

### Official Review · Reviewer_7qYH · 2025-11-01

**Soundness:** 2
**Presentation:** 2
**Contribution:** 3
**Rating:** 2
**Confidence:** 4

**Summary:**

This paper introduces Layer Query Network (LQN): a five-layer MLP that performs layer-wise spatial 3D coordinate querying and distillation for a frozen VLM under single-sample Test-Time Training (TTT). By employing Binding and Recirculation (path consistency) losses, it achieves efficient adaptation without relying on heavy data augmentation or backpropagation through large backbones. The authors report superior performance compared to various TTT/TTPrompt methods on tasks such as natural distribution shifts, cross-dataset classification, and semantic/instance/panoptic segmentation, while also showing better GFLOPs efficiency in several settings.

**Strengths:**

1.	The experiments are quite comprehensive, covering both classification tasks and segmentation tasks.
2.	The issue of non-convergence is avoided by using positional encoding.

**Weaknesses:**

1.	The writing is not very clear. Does "At every iteration" in line 164 refer to each batch rather than every iteration during student training? Figure 2 has not been referenced, which makes it necessary to go back and check when (src, dest) is mentioned in line 215.

2.	The motivation is still not clear. Why does positional encoding prevent non-convergence in this problem? And why is the Recirculation process designed to enforce spatial invariance? Can changing the source position maintain spatial consistency?

3.	The method section is not detailed enough. How is P specifically implemented?

4.	The introduction of innovation is insufficient. Is it just a modification of the APM method by applying multiple layers of distillation? The new challenges faced and the motivation for the proposed solutions are not fully explained.

**Questions:**

1. Sensitivity analysis experiments on the same dataset are necessary. Which dataset is the reported α based on in the paper? Does it vary across different datasets?
2. If 5% of the samples are randomly selected, is the random selection of parameters important? Will different 5% samples cause significant differences?
3. Have you tried other VLM models besides CLIP?

---

> ### Author Response · Authors · 2025-11-19
> **Authors response to the respected reviewer Reviewer 7qYH31 (1/3)**
>
> We thank the respected reviewer for helping us improve our work. Please find our comments as below:
>
> `W1  On clarifications of "iterations" in line 164, referencing fig 2 in line 215`
>
> We apologize for the lack of clarity on our part. For a particular test sample, we perform $15$ iterations where the LQN-student mimics teacher features. The weights are reset after these $N$ iterations to prevent information leakage between multiple test samples.
>
>
> We appreciate the observation and will fix the missing reference to Figure 2. We will also transfer it to page 4 so that a reader does not have to scroll back and forth.
>
> `W2.1  On motivation, and how convergence is resolved by positional encodings`
>
> Our motivation behind designing LQN's is to enable:
> VLM's system to adapt to OOD samples, without having to tune all of its parameters
> Explore adaptation under data-constrained scenarios, when as few as 1 test sample may be available (TTT).
>
> Positional encodings play a crucial role in stabilizing optimization by encoding location-dependent information directly into the feature representation. Without positional cues, features extracted from different spatial locations can become ambiguous or interchangeable, making it difficult for the model to align and propagate gradients consistently across layers. By embedding explicit positional information, the network learns to distinguish spatially distinct representations (e.g., regions L1​ and L2 of the same image), ensuring that learning updates at each spatial coordinate remain well-conditioned. This facilitates convergence even in shallow or mid-level layers, rather than relying solely on high-level representations. Moreover, Nerf [5] showed that using positional embedding helps small networks mimic RGB pixels faithfully. We are applying this principle in mimicking VLM features via 3D positional embeddings.
>
>
> `W2.2  On reasons behind enforcing spatial-invariance `
>
> Because LQN processes each location independently, we need a mechanism for locations to communicate across spatial locations, so that global spatial consistency is preserved. In classical self-attention, this is handled by a dense attention mask interaction graph: each patch attends to every other patch (and incurs quadratic cost).
>
> LQN processes locations in parallel. There needs to be a constraint where a pair of locations are able to interact (couple) with each other. Our recirculation mechanism is a way to provide this coupling. Conceptually similar ideas have appeared in earlier theoretical work on energy-based and recurrent models, such as the recirculation algorithm [2], but our design adapts this principle to the modern TTT / VLM setting.
>
>
>
>
>
> `W2.3 On effects of fluctuating source during recirculation`:
>
> Thanks for the deep insight. For a particular destination, there is only one answer (which we distill from the teacher). Therefore, when different sources interact, they should still predict the same value. That is the core inspiration of recirculation, and we enforce it as a constraint on the net on Lines 13-20 in Algorithm 1. In fig 6 (iii) without such consistency LQN only obtains $69.4 $. With consistency, LQN gets $70.2$, which indicates an improvement of $0.8$. We will make it clearer in the draft.
>
> `W3 On clarifications on implementation of P`
>
> We apologize for the lack of clarity on our part. We will modify lines 178-179 of the paper to "We implement P as a 3D positional-encoding in transformers [3]". We will further include the mathematical equations governing P (similar to sec 3.5 of transformer paper [3]).
>
> `W4 On the  subtleties of lqn vs apm`:
>
> LQN is able to perform intermediate layers distillation using 3D positional embedding. We draw inspiration from Nerf [5] that small networks can mimic RGB pixels with the help of positional embeddings. We use this to mimic VLM features at a desired spatial / depth location, making it work on tasks beyond classification, i.e., image segmentation. APM is focused on classification tasks, and since it only distills global features, it can not model the internal features of the teacher. Modelling intermediate (internal) teacher representations leads to a new inductive bias: the ability to access internal representations in a constant-time (line 428 of paper) as compared to O(L) time in classical neural nets [5].

---

> > ### Comment · Reviewer_7qYH · 2025-11-28
> >
> > I appreciate the authors' rebuttal, which has addressed most of my concerns. At this stage, I intend to maintain my original score and would be interested in seeing the feedback from the other reviewers.

---

> ### Author Response · Authors · 2025-11-19
> **Authors response to the respected reviewer Reviewer 7qYH31 (2/3)**
>
> `Q1 On alpha, it's values across different datasets, 5\% location selection for recirculation`
>
> We apologize for not reporting the hyperparameter variation across datasets.
> The original alpha =  0.7 was reported for the ImageNet val set. For other datasets, the alpha varies within +-0.3 standard deviation around 0.7. For eg, alpha is 0.6 for ImageNet-A, ImageNet-v2, Pets, Sun397, Eurosat. Similarly, alpha is  0.8 for ImageNet-R, UCF101, Food101. For all other datasets, it is 0.7. We will update the appendix Sec 8 with these numbers.  We have also provided additional analysis of the impact of varying alpha over ttt iterations in
>
> | Dataset       | Alpha |
> |---------------|----------------|
> | ImageNet      | 0.7           |
> | Flowers102    | 0.5           |
> | DTD           | 0.7           |
> | Caltech101    | 0.5           |
> | Aircraft      | 0.7           |
>
>
> `Q2 5% sampling`
>
> Additionally, we perform experiments on the ImageNet val set for multiple runs, sampling `different' $5\%$ locations every time and report the mean and standard deviation. We observe the following results:
>
> | Model | Run 1 | Run 2 | Run 3 | Run 4 | Run 5 | Mean  | Std  |
> |-------|-------|-------|-------|-------|-------|-------|------|
> | LQN   | 70.11 | 70.40 | 69.90 | 70.20 | 70.30 | 70.10 | 0.17 |
>
> The low deviation $0.17$ indicates the relative stability of LQN against randomly sampling.  Intuitively, when two locations contain different semantic information, it should help the learning process.  One interesting variation we noticed was when these locations are sampled from objects belonging to different locations. Specifically, we ran an object detector (EomT) on input image to isolate individual objects, and sample locations from different objects.  This increases the performance from 70.2 to 71.5. However, the presence of an additional object detector makes the TTT computationally really expensive.
>
> `Q3 On other models besides clip`
>
> We provide more results of LQN w.r.t. other VLMs like SigLIP, EVA, COCA. We will add these to the main paper.
>
> ### SigLIP Results
> | Method      | ImageNet | ImageNet-A | ImageNet-V2 | ImageNet-R | ImageNet-Sketch | Avg   | OOD Avg |
> |-------------|----------|------------|-------------|------------|------------------|--------|---------|
> | SigLIP      | 76.0     | 45.3       | 68.9        | 90.3       | 67.9             | 69.68  | 68.10   |
> | **LQN**     | **78.3** | **47.0**   | **71.1**    | **92.6**   | **70.0**         | **71.80** | **70.18** |
>
> ---
>
> ### EVA-CLIP Results
> | Method      | ImageNet | ImageNet-A | ImageNet-V2 | ImageNet-R | ImageNet-Sketch | Avg   | OOD Avg |
> |-------------|----------|------------|-------------|------------|------------------|--------|---------|
> | EVA-CLIP    | 76.1     | 64.6       | 68.9        | 89.1       | 63.3             | 72.40  | 71.48   |
> | **LQN**     | **78.1** | **66.8**   | **71.0**    | **91.8**   | **65.7**         | **74.68** | **73.83** |
>
> ---
>
> ### COCA Results
> | Method      | ImageNet | ImageNet-A | ImageNet-V2 | ImageNet-R | ImageNet-Sketch | Avg   | OOD Avg |
> |-------------|----------|------------|-------------|------------|------------------|--------|---------|
> | COCA        | 63.6     | 21.5       | 55.7        | 73.2       | 51.3             | 53.06  | 50.43   |
> | **LQN**     | **65.7** | **22.9**   | **57.7**    | **75.5**   | **53.3**         | **55.02** | **52.35** |
>
>
>
> ### SigLIP Results
> | Method    | Flowers102 | DTD  | Pets | UCF101 | Caltech101 | Food101 | SUN397 | Aircraft | EuroSAT | Avg  |
> |-----------|------------|------|------|--------|-------------|---------|--------|----------|---------|------|
> | SigLIP    | 85.8       | 64.7 | 94.1 | 72.5   | 90.5        | 89.8    | 69.8   | 43.8     | 43.8    | 72.7 |
> | **LQN**   | **87.8**   | **66.7** | **95.8** | **74.4** | **92.6** | **91.5** | **71.2** | **45.9** | **45.5** | **74.6** |
>
>
> ### COCA Results
> | Method    | Flowers102 | DTD  | Pets | UCF101 | Caltech101 | Food101 | SUN397 | Aircraft | EuroSAT | Avg  |
> |-----------|------------|------|------|--------|-------------|---------|--------|----------|---------|------|
> | COCA      | 64.7       | 53.3 | 89.1 | 61.4   | 89.1        | 77.3    | 66.1   | 18.8     | 45.3    | 62.7 |
> | **LQN**   | **66.8**   | **55.7** | **91.5** | **63.8** | **91.0** | **79.9** | **68.6** | **20.6** | **47.3** | **65.0** |
>
>
> ### EVA-CLIP Results
> | Method    | Flowers102 | DTD  | Pets | UCF101 | Caltech101 | Food101 | SUN397 | Aircraft | EuroSAT | Avg  |
> |-----------|------------|------|------|--------|-------------|---------|--------|----------|---------|------|
> | EVA-CLIP  | 72.0       | 59.3 | 93.7 | 74.1   | 90.4        | 89.7    | 71.9   | 28.5     | 69.9    | 72.1 |
> | **LQN**   | **74.2**   | **61.4** | **95.7** | **76.4** | **92.6** | **91.5** | **74.0** | **30.5** | **72.3** | **74.2** |
>
>
> We will be happy to respond to further clarifications. Thank you so much for your valuable time,
>
> Yours Sincerely,
>
> Authors

---

> > ### Author Response · Authors · 2025-11-19
> > **Authors response to the respected reviewer Reviewer 7qYH31 (3/3)**
> >
> > P.S. References used in the rebuttal:
> >
> > [1] Hinton, Geoffrey E. "Relaxation and its role in vision." (1977).
> >
> > [2] Hinton, Geoffrey E., and James McClelland. "Learning representations by recirculation." Neural information processing systems. 1987
> >
> > [3] Vaswani, Ashish, et al. "Attention is all you need." Advances in neural information processing systems 30 (2017).
> >
> > [4]https://github.com/tatp22/multidim-positional-encoding/blob/efeb8d9d70e8184da50eae9fddd1bbda10896529/positional_encodings/torch_encodings.py#L142
> >
> > [5] Mildenhall, Ben, et al. "Nerf: Representing scenes as neural radiance fields for view synthesis." Communications of the ACM 65.1 (2021): 99-106.
> >
> > [6] Bengio, Yoshua, et al. "Greedy layer-wise training of deep networks." Advances in neural information processing systems 19 (2006).

---

> ### Author Response · Authors · 2025-11-30
> **Official Comment by Authors**
>
> Dear Reviewer 7qYH,
>
> Thank you for the thoughtful and constructive feedback. In response to all the reviewers' comments, we have substantially expanded our analysis, including comparisons with MTA and GS-Bias, quantitative evaluation of texture-based image generation, additional generalization on other VLMs, further ablations, and extensions to test-time adaptation.
>
> If there are specific aspects that the reviewer feel could further strengthen the contribution or improve the clarity of the work, we would be grateful for your guidance and would be happy to incorporate additional improvements.
>
> Sincerely,
>
> Authors

---

### Official Review · Reviewer_ujEh · 2025-11-01

**Soundness:** 2
**Presentation:** 3
**Contribution:** 2
**Rating:** 4
**Confidence:** 1

**Summary:**

To deal with OOD issue for CLIP-like models, previous solutions are finetuning, Test-Time Adaptation, or Test-Time Training. In this paper, authors explore if we really need finetuning and proposed e Layer Query Network, a lightweight module that can be injected to a forzen CLIP model to improve the generalization ability. Extensive experiments on various benchmarks show great performance.

**Strengths:**

1. Good presentation makes the paper easy to follow.
2. Extensive experiments show promising performance.

**Weaknesses:**

1. If the 3D positional embedding is just learnable parameters for 2D spatial information and layer index?

2. I have a question about the supervision of intermidiate features. based on empirical experience, Distialltion with guidance on intermidiate features cannot provide significant improvements. If any results that without L_B in Algorithm 1?

3. Im not sure if MSE loss for Eq (3) and Eq (4) is good choice.

**Questions:**

1. It would be a little misleading in the title "VISION-LANGUAGE-MODELS", people would consider large multimodal language models.

2. Authors claimed "Faster convergence" in Line 84, if any experimental results support that?

---

> ### Author Response · Authors · 2025-11-19
> **Authors response to the respected reviewer ujEh**
>
> We thank the respected reviewer for helping us improve our work. Please find our comments as below:
>
> `W1 on clarifications of learnable/fixed-form of  3D positional encodings `
>
> We would be happy to clarify that 3D positional encodings used in our work are sinusoid and cosine encodings from the transformer paper [1]. They do not contain **any learnable parameters**. We will add this clarification to the paper. Trainable 3D Positional embedding (H x W x D, e.g., 56 x 56 x 12) is infeasible to train in our setting of 1 test sample trained over 15 iterations of forward/backward passes.
>
>
> `W2 on intermediate feature-distillation, effects without binding loss $L_b$ `
>
> We thank the reviewer for the deep insight. In our model design, a small LQN network starts from random weights for every test sample, learns to mimic `all the intermediate features` of the teacher VLM via $L_B$ binding loss. Without it, LQN does not know what features to mimic. The `final MLP layers` layer mimicking both the shallow and deep features of the VLM, improves the performance on dense tasks as a whole with faster convergence.
>  .
>
> `W3 Plausibility of existence of different losses better than MSE`
>
> We agree with the reviewer that better losses than MSE could be used. In LQN, MSE loss outperformed entropy loss over augmented test-samples (lines 476-477), and is **primarily a hyperparameter / design choice**. Indeed, there could be a better loss formulation. We tested TTT on LQN for ImageNet with two additional losses (L1 loss and cosine similarity). We note the following results:
>
> ### LQN Pretraining Loss Comparison (ImageNet)
>
> | Method                      | ImageNet |
> |-----------------------------|---------------|
> | LQN (L1 Loss)          | 68.6          |
> | LQN (MSE Loss)      | 70.2          |
> | LQN (Cosine Loss)   | 71.4          |
>
> Indeed, cosine loss performs better than MSE loss. We will add this analysis in the revised manuscript.
>
> `Q1 on  VLM in title being possibly mis-leading`
>
> Thank you for pointing this out. Our work adapts CLIP and other models like CLIP, i.e. SigLIP, Eva-CLIP, and Coca, which are commonly referred to as Vision Language models. The performance improvement is largely due to distilling these VLMs; hence, the entire system as a whole improves the performance. Similar examples in literature exists, like TPS [4] *“Just shift it: Test-time prototype shifting for zero-shot generalization with **vision-language models**”* actually only updates the text feature (no VLM / backbone involved), and TPT [5] *"Test-time prompt tuning for zero-shot generalization in **vision-language models**"*, modifies trainable prompts without modifying VLMs.
>
>
> `Q2 on faster-convergence claims`
>
> We compare the performance of LQN with other methods in Fig 4 (left).  LQN only consumes 15 iterations, whereas other methods (APM) consume $20$ iterations.Similarly, LQN obtains better results and lower time (47min) than other methods (TPT takes 9h 15min), (DiffTPT takes 20h), (TPS takes 55min). Therefore, we observe faster-convergence.
>
>
> We will be happy to respond to further clarifications. Thank you so much for your valuable time,
>
> Yours Sincerely,
>
> Authors
>
>
>
> [1] Vaswani, Ashish, et al. "Attention is all you need." Advances in neural information processing systems 30 (2017).
>
> [2] Sitzmann, Vincent, et al. "Implicit neural representations with periodic activation functions." Advances in neural information processing systems 33 (2020): 7462-7473.
>
> [3] Carion, Nicolas, et al. "End-to-end object detection with transformers." European conference on computer vision. Cham: Springer International Publishing, 2020.
>
> [4] Sui, Elaine er al. “Just shift it: Test-time prototype shifting for zero-shot generalization with vision-language models”. IEEE/CVF Winter Conference on Applications of Computer Vision (WACV), pp. 825–835. IEEE, 2025.
>
> [5] Shu, Manli, et al. "Test-time prompt tuning for zero-shot generalization in vision-language models." Advances in Neural Information Processing Systems 35 (2022): 14274-14289.

---

> > ### Comment · Reviewer_ujEh · 2025-11-28
> >
> > Thanks for the rebuttal, which addressed most of my concerns.
> >
> > I'm more interested in the LQN Pretraining Loss Comparison (ImageNet) experiments.
> >
> > Could you please provide any insights why cosine performs best, and L1 performs worst (we can see clear performance gap from difference losses).

---

> > > ### Author Response · Authors · 2025-11-30
> > >
> > > We thank the reviewer for their insightful questions on ImageNet experiments.
> > >
> > > Please find some of our insights below:
> > >
> > > - Consider two vectors, one being predicted by the the LQN student (x) and one by the LQN teacher (y). Cosine promotes angular similarity between these vectors and not necessarily norm (it does not need to match the magnitude as long as it can match direction). There are papers like CLIP which have observed that  matching directions is sufficient.
> > >
> > > - L1 promotes weight-sparsity, a form of regularization. Small LQN MLP is supposed to distill (mimic) CLIP image-encoder while being 17x smaller than VIT-B/16's. This is a classic case of underfitting and L1 loss on top, promoting sparsity, makes small models job even harder. L2 loss in contrast, does not have this regularization effect.

---

### Author Response · Authors · 2025-11-19
**Authors global response to all the respected reviewers.**

Respected Reviewers,

We appreciate the positive feedback from the reviewers for our work. The reviewers acknowledged several aspects such as good presentation `[Reviewer ujEh, cDhr]`, extensive-experiments `[Reviewer cDhr, 7qYH, 83JT]`, convincing ablations`[Reviewer 83JT]`, avoiding non-convergence issues via positional-encodings `[Reviewer 7qYH]`. Based on reviewers constructive-feedback, we make further positive-amendments to our work;

- We update our paper, with discussions in related work (MTA, GS-Bias)`[Reviewer cDhr]`, adding more quantitative comparisons in Tables 1, 2`[Reviewer cDhr]`.
- We provide additional analysis on challenging images with repeating textures in Fig 10`[Reviewer cDhr]`.
- We evaluate LQN on additional VLMs like Sig-CLIP, Eva-CLIP and COCA`[Reviewer 83JT,Reviewer 7qYH]`.
- We discuss ablations on varying the number of sampled src locations[Reviewer 7qYH,Reviewer 83JT] , and provide dataset-specific hyperparameters`[Reviewer 7qYH]`.
- We evaluate additional baselines where LQN is allowed to retain information across different test-samples, and study the accuracy-time tradeoff `[ Reviewer 83JT]`.
- We have uploaded the revised draft in the openreview portal, with changes clearly marked  in blue. We welcome your further feedback.

`Addressing paper-writing comments`
- We have fixed some writing issues (captions on top of tables) `[ Reviewer 83JT]`.
- We will further  clarify the motivations of LQN`[Reviewer 7qYH]`,difference in distillation process compared to APM`[Reviewer 7qYH]`,  add citations to fig 2`[Reviewer 7qYH]`, move algorithm 1 to supplementary `[Reviewer 83JT]`, clarify positional encodings implementation in paper `[Reviewer 7qYH]` ,

We thank the reviewers for giving our work their valuable time. We have also provided individual-responses and look forward to the chance to engage in further-discussions.

Yours Sincerely,

Authors.

---

### Note · Program_Chairs · 2026-01-17
**Submission Desk Rejected by Program Chairs**

The following references in this submission do not refer to real documents and/or have major errors in bibliographic information:

     Dan Hendrycks, Kevin Zhao, Steven Basart, Jacob Steinhardt, and Dawn Song. Imagenet-r: Background challenge for imagenet. In Advances in Neural Information Processing Systems (NeurIPS), 2021b.